



# The regional impact of urban emissions on air quality in Europe: the role of the urban canopy effects

Peter Huszar[1], Jan Karlický[1,3], Jana Marková[1,2], Tereza Nováková[1], Marina Liaskoni[1], and Lukáš Bartík[1]

[1]Department of Atmospheric Physics, Faculty of Mathematics and Physics, Charles University, Prague, V Holešovičkách 2, 18000, Prague 8, Czech Republic
[2]Czech Hydrometeorological Institute (CHMI), Na Šabatce 17, 14306, Prague 4, Czech Republic
[3]Institute of Meteorology and Climatology, Department of Water, Atmosphere and Environment, University of Natural Resources and Life Sciences, Vienna, Gregor-Mendel-Straße 33, 1180 Vienna, Austria

**Correspondence:** P. Huszar (huszarpet@gmail.com)

**Abstract.** Urban areas are hot-spots of intense emissions and they influence air-quality not only locally but on regional or even global scales. The impact of urban emissions over different scales depends on the dilution and chemical transformation of the urban plumes which are governed by the local and regional scale meteorological conditions. These are influenced by the presence of urbanized land-surface via the so called urban canopy meteorological forcing (UCMF). In this study, we investigate

for selected central European cities (Berlin, Budapest, Munich, Prague, Vienna and Warsaw), how the urban emission impact (UEI) is modulated by the UCMF for present day climate conditions (2015-2016) using three regional climate-chemistry models: the regional climate models RegCM and WRF-Chem (its meteorological part), the chemistry transport model CAMx coupled to either RegCM and WRF and the "chemical" component of WRF-Chem. The UCMF was calculated by replacing the urbanized surface by rural one while the UEI was estimated by removing all anthropogenic emissions from the selected

cities.

We analyzed the urban emissions induced changes of near surface concentrations of $NO_2$, $O_3$ and PM2.5. We found increases of $NO_2$ and PM2.5 concentrations over cities by 4-6 ppbv, and 4-6 $\mu gm^{-3}$, respectively meaning that about 40-60% and 20-40% of urban concentrations of $NO_2$ and PM2.5 are caused by local emissions and the rest is the result of emissions from the surrounding rural areas. We showed that if UCMF is included, the UEI of these pollutants is about 40-60% smaller, or

in other words, the urban emission impact is overestimated if urban canopy effects are not taken into account. In case of ozone, models due to UEI usually predict decreases around -2 to -4 ppbv (about 10-20%), which is again smaller if UCMF is considered (by about 60%). We further showed that the impact on extreme (95th percentile) air-pollution is much stronger, as well as the modulation of UEI is larger for such situations. Finally, we evaluated the contribution of the urbanization induced modifications of vertical eddy-diffusion to the modulation of UEI, and found that it alone is able to explain the modelled decrease of the

urban emission impact if the effects of UCMF are considered. In summary, our results showed that the meteorological changes resulting from urbanization have to be included in regional model studies if they intend to quantify the regional fingerprint of urban emissions. Ignoring these meteorological changes can lead to strong overestimation of UEI.



## 1 Introduction

Already more than 50% of the human population lives in urban areas and an increase over 60% during the upcoming decades
is foreseen (UN, 2018). The consequences of urbanization (i.e. the transition from rural to urban surfaces) on atmospheric
conditions are evident (Folberth et al., 2015) and they affect both the climate (Zhao et al., 2014; Zhu et al., 2017; Huszar et al.,
2014; Karlický et al., 2018; Karlický et al., 2020) and air pollution (Freney et al., 2014; Timothy and Lawrence, 2009; Butler
and Lawrence, 2009; Im and Kanakidou, 2012; Huszar et al., 2016a), and the possible interactions between them (e.g. Huszar
et al., 2018b; Han et al., 2020; Huszar et al., 2020a; Fan et al., 2020) that often lead to complex counteracting effects (Yu et al.,
2020).

In principle, cities influence the physical and chemical state of the atmosphere via two primary pathways. First of all, urban
canopies are covered by artificial materials and objects in a specific geometric layout (building and streets) resulting in a range
of effects on the meteorological conditions – i.e. they act via the so called urban canopy meteorological forcing (UCMF) as
defined by Huszar et al. (2020a). In particular, temperature is increased in cities due to the urban heat island effect (Oke,
1982; Oke et al., 2017). Due to enhanced roughness, the city scale wind speed is decreased (Huszar et al., 2014; Jacobson
et al., 2015; Zha et al., 2019) while for turbulence (especially the vertical eddy diffusivity), strong increase is seen (Barnes
et al., 2014; Huszar et al., 2018b; Ren et al., 2019; Huszar et al., 2020a). Secondly, due to high population density and thus
concentrated human activities and energy demand, cities represent intense source of both greenhouse (Folberth et al., 2012)
and short lived pollutants that impact not only the local air quality but act over regional scales (Freney et al., 2014; Panagi et
al., 2020) or even global ones (Butler and Lawrence, 2009).

Indeed, cities emit large quantities of different pollutants with various chemical characteristics. They encompass the emissions of oxides of nitrogen (NOx) emitted mainly by road transportation along with non-methane volatile organic compounds (NMVOCs). Depending on their ratio, the photochemical regime in and around cities is determined being either
NOx-controlled or VOC-controlled (Xue et al., 2014). The ratio NOx/VOC is in general high in North-American urban
agglomerations, eastern Asian cities and in European megacities like Paris, Milan or Athens as well and ozone is predominantly titrated over these cities (Beekmann and Vautard, 2010). In such cities, emission controls to reduce pollution often face
counteracting effect when reduced NOx and NMVOC emissions lead to ozone increase, as seen recently in many urban areas
due to the COVID-19 pandemic induced traffic reductions (Salma et al., 2021; Lamprecht et al., 2021; Putaud et al., 2021;
Grange et al., 2021) or shown previously also by Huszar et al. (2016b).

Carbon monoxide (CO) and methane ($CH_4$) play a rather minor role in ozone production over and around cities, however,
CO remains important for its harmful effect on human health (Bascom et al., 1996) and turned also out to be a good tracer to
identify the sources of urban pollution (Panagi et al., 2020).

Emissions of gaseous pollutants further perturb aerosol concentration. In presence of water droplets, emissions of NOx,
sulfur dioxide ($SO_2$) and ammonia ($NH_3$) lead to formation of secondary inorganic aerosol (SIA). The primary precursor
for sulfate aerosol ($PSO_4$) formation is $SO_2$. Although $SO_2$ emissions over the last decades are decreasing globally (Zhong
et al., 2020), significant perturbation of aerosol burden is found in many urbanized regions (Guttikunda et al., 2003; Yang





et al., 20110(@). Apart from affecting photochemistry, emissions of NOx lead to formation of nitrate aerosol (PNO$_3$). If the meteorological conditions are favorable, NOx from cities can enhance background PNO3 levels significantly (Lin et al., 2010). Emissions of ammonia(NH$_3$), although not emitted largely by cities, are an efficient contributor to formation of sulfate and

nitrate aerosol (by forming ammonium-sulfates and ammonium-nitrates) and its importance in connection with city emissions is highlighted by many (e.g. Behera and Sharma, 2010, and references therein). In general, the thermodynamic system of ammonium-sulfate-nitrate-water solution is rather complicated and its equilibrium state is highly dependent on the initial ratio (emission) of SO$_2$-NOx-NH$_3$ and the prevailing meteorological conditions (Martin et al., 2004), thus high variability in the contribution of different cities to aerosol is expected. Apart from the SIA, directly emitted organic and elemental carbon

(OC/EC) can be also a major fraction of the urban aerosol impact, as shown e.g. for Paris by Freney et al. (2014).

As the dilution of urban plumes into larger scale includes its mixing with rural emissions and formation of secondary pollutants, its atmospheric fingerprint requires complex modeling experiments where both gas phase and aerosol chemistry and transport is simultaneously considered and coupled to meteorological conditions. Indeed, numerous modeling studies attempted to evaluate the impact of urban emissions at different scales. On global scale, e.g. Lawrence et al. (2007), Butler

and Lawrence (2009), Folberth et al. (2010) or Stock et al. (2013) estimated urban emissions impact, while on regional scales, many studies focused on agglomerations in southern Europe (e.g. Im et al., 2011a, b; Im and Kanakidou, 2012; Finardi et al., 2014), but focused also on other important urban centers like Paris (Skyllakou et al., 2014; Markakis et al., 2015) or London (Hodneborg et al., 2011; Hood et al., 2018). Huszar et al. (2016a) showed for multiple cities in central Europe that although air pollution in cities is determined mainly by the local sources, significant (often a tens of %) fraction of the concentration is

associated to other sources from rural areas and minor cities. There have been also studies that investigated strongly polluted eastern Asian cities (Guttikunda et al., 2003, 2005; Tie et al., 2013).

When investigating the impact of a particular source or source region on air quality, many approaches are available while for urban emissions, usually either the annihilation method (Baklanov et al., 2016; Huszar et al., 2016a) was applied which means comparing experiments with and without the source emissions flux or the tracer approach which marks the source

region with a less reactive or inert tracer and tracks its dispersion as done for e.g. CO as tracer recently in (Panagi et al., 2020). In any of the cases, two aspects are important to consider: i) it has to be ensured that the non-linear chemical effects during dispersion onto the resolved model scale are taken into account. This was widely considered in case of emissions from different transport modes (Huszar et al., 2010, 2013), for emission from urban areas however its importance is probably relatively minor (Markakis et al., 2015). Furthermore ii) the meteorological conditions responsible for the initial dilution and

dispersion of the urban plume have to be correctly captured. In case of urban emissions this is especially crucial as over urban areas, meteorological conditions are significantly perturbed by the characteristics of the urban canopy while virtually all meteorological parameters are perturbed (Karlický et al., 2020). Indeed, many studies looked at urban air quality from the perspective of the influence of the urban canopy and found large perturbations of the absolute values of NOx, O$_3$ and particle matter (PM) while the impact of turbulence, wind and temperature modifications showed to be the most important (Wang et

al., 2007; Struzewska and Kaminski, 2012; Liao et al., 2014; Kim et al., 2015; Zhu et al., 2017; Zhong et al., 2018; Li et al., 2019; Huszar et al., 2018a, b, 2020b) leading together to decreases of primary pollutants and increases in ozone or e.g.



secondary organic aerosol (Huszar et al., 2018a; Janssen et al., 2017). Recently, Ulpiani (2021) argued too that the urban heat island (UHI) and the urban pollution island (UPI) have to be assessed in a common framework as the governing physical and chemical mechanisms are strongly linked. In other words, the local and regional fingerprints of urban emissions are strongly influenced by the weather conditions in and around the particular city.

It is thus clear that when evaluating the urban emission impact, the effect of the urban canopy meteorological forcing has to be taken into account as it is strongly probable, that the UCMF has a significant modulating effect. The first family of listed studies that modeled the urban emission impact in the last decade did not include these canopy effects. On the other hand, studies that dealt with the impact of the UCMF on air quality had to, in principle, include urban meteorological effects into account, however, they did not explicitly focused on the impact of emissions but they looked at the absolute concentrations influenced by both the background air pollution and the input from the particular urban area. Here we propose a study that connects the two aspects of anthropogenic modifications of urban atmosphere, i.e. the impact of urban emissions and the impact of urban canopy on meteorological conditions. The study explicitly asks and evaluates, what is the contribution of the UCMF to the urban emission impact (UEI) or in other words how the magnitude of UEI depends on the (non)inclusion of UCMF. To evaluate this we will adopt a multi-model approach on regional domain for present day conditions and perform multiple experiments differing in including/excluding both the effect of UCMF and the impact of the urban emissions. Special attention will be dedicated to the importance of vertical eddy transport as it is believed to be an important or even dominating driver of the regional fingerprint of urban emissions (see e.g. Huszar et al. (2020a)). The analyzed species are ozone ($O_3$), nitrogen dioxide ($NO_2$) and particle matter with diameter less then 2.5 $\mu$m (PM2.5). Cities considered regarding the urban emissions they emit are large central European metropoles: Prague, Berlin, Munich, Budapest, Vienna and Warsaw.

The paper consists of four main parts: after the Introduction, the models, their configuration, the experiments and the data implemented are described in the Methodology section. In the Results section, the results are presented which include the evaluation of the urban emission impact and how this impact is modulated by the UCMF, this also involves the presentation of the impact on extreme air pollution values and finally, the impact of the turbulence alone on the total emission impact is analyzed. Finally, the results are discussed and conclusions are drawn.

## 2 Methodology

### 2.1 Models used

Two models regional climate models (RCM) as meteorological drivers, RegCM version 4.7 and WRF-Chem version 4.0.3 and two chemical transport models (CTM), the CAMx version 6.50 and the online coupled chemical module of the WRF-Chem model were used in the study. As the models and their parameterizations are identical to those in Huszar et al. (2020b), here we will provide only the most relevant information.

RegCM4.7 is a non-hydrostatic limited-area climate model described in (Giorgi et al., 2012). Boundary layer physics, cloud/rain micro-physics and convection were treated by the Holtslag PBL parameterization (HOL; Holtslag et al., 1990), WSM5 5-class moisture scheme (Hong et al., 2004) and the Tiedtke scheme (Tiedtke et al., 1989). The meteorological phe-





nomenon associated with urbanized surfaces was taken into account using the CLMU urban canopy module implemented in the CLM4.5 (Oleson et al., 2008, 2010, 2013) land-surface scheme. CLMU represents cities in the classical canyon geometry. To calculate the heat and momentum fluxes in the urban canyon, Monin-Obukhov similarity theory with roughness lengths and displacement heights typical for the canyon environment is invoked (Oleson et al., 2010). Anthropogenic heat flux from air conditioning and heating is computed based on (Oleson et al., 2008).

WRF-Chem is a regional weather and climate model including chemistry described in Grell et al. (2005). In our setup, the Purdue Lin scheme (Chen and Sun, 2002, PLIN;) for microphysics, the BouLac PBL scheme (Bougeault and Lacarrère, 1989), the Grell 3D convection scheme (Grell, 1993) and the Single-Layer Urban Canopy Model (SLUCM; (Kusaka et al., 2001)) to account for the urban canopy meteorological effects are used. Online coupled to the main meteorological part, the chemical module of WRF-Chem invokes here the gas-phase Regional Acid Deposition Model, v. 2 (RADM2; Stockwell et al.

(1990, 2011)) mechanism and the Modal Aerosol Dynamics Model for Europe and Secondary Organic Aerosol Model module (MADE/SORGAM; Schell et al. (2001)) scheme for aerosol.

Another model for the chemistry simulations is the chemistry transport model CAMx version 6.50 (ENVIRON, 2018). CAMx is a photochemical CTM working in Eulerian framework and implements multiple gas phase chemistry schemes (Carbon Bond 5 and 6, SAPRC07TC) with the Carbon Bond 5 (CB5) scheme (Yarwood et al., 2005) used in this study. Static two

mode approach is considered for particle matter. For secondary inorganic aerosol, the ISORROPIA thermodynamic equilibrium model (Nenes and Pandis, 1998) is activated. Secondary organic aerosol (SOA) concentrations are calculated using the SOAP equilibrium scheme (Strader et al., 1999). Dry and wet deposition are solved with the Zhang et al. (2003) and Seinfeld and Pandis (1998) methods, respectively.

Meteorological driving data for CAMx are taken either from the RegCM model or from WRF-Chem (i.e. its atmospheric

part). Meteorological preprocessor is used to translate the RegCM and WRF meteorological data in to model-ready driving data for CAMx: the wrfcamx preprocessor supplied along with the CAMx code http://www.camx.com/download/support-software. aspx was used for WRF data while for RegCM, the RegCM2CAMx interface originally developed by Huszar et al. (2012) was applied. The vertical eddy diffusion coefficients ($K_v$) are calculated using the CMAQ diagnostic approach (Byun, 1999). Given the fact, that the coupling between CAMx and the driving models is offline, no feedbacks of the species concentrations

on WRF/RegCM radiation/microphysical processes were considered. Indeed, Huszar et al. (2016b) showed that their long term effect is very small.

## 2.2 Model setup, data and simulations

Model simulations were performed over the same nested domains and for the same period as in Huszar et al. (2020b) with 9km, 3km and 1km resolution centered over Prague, Czechia (50.075° N, 14.44° E; Lambert Conic Conformal projection).

The model orography including the placement of the three domains and the cities analyzed are presented in Fig.1. The model grid spawns 40 layers in vertical in both RegCM and WRF-Chem. The thickness of the lowermost layer is about 30 m and the model top is at 5 hPa (corresponding to about 36 km). The simulated time-period is 2014 Dec – 2017 Jan period with the first month used as spin-up. As for the resolution, according to Tie et al. (2010), the threshold for the ratio of size of the analyzed





city to resolution should be around 1:6, which means 6 km or higher spatial resolution should be used to assess the emission
impact of the cities we will focus. For Prague analyzed at 1 km, this is fulfilled, for other cities outsides of the inner 1 km
domain and usually outside of the middle 3 km domain, the resolution is somewhat coarser but we will rely on the findings of
studies that looked at the impact of resolution on the species concentration and found that the impact is rather small (Hodneborg
et al., 2011; Markakis et al., 2015; Huszar et al., 2020a). Wang et al. (2021) recently showed for the case of Hong Kong, that
ozone production is reduced if high resolution is applied (Large-Eddy Simulation) but the decrease is small (around 8% for
near surface ozone concentrations). Even here we will later see that the city scale impact for Prague is similar between the 9
and 1 km resolution.

For the coarse 9 km domain simulations,the ERA-interim reanalysis (Simmons et al., 2010) is used as climate forcer. The 3
and 1 km domains are forced by the corresponding parent domains with one-way nesting. Chemical boundary conditions (for
the outer domain) are taken from the CAM-chem global model data (Buchholz et al., 2019; Emmons et al., 2020)). Landuse
information adopted in model simulations was derived from the high resolution (100 m) CORINE CLC 2012 landcover data ()
and the United States Geological Survey (USGS) database for gridcells without CORINE information. An important difference
between WRF and RegCM models is that the latter one, fractional landuse is considered while in WRF, each gridcell is
attributed the dominant landuse.

### 2.2.1 Model simulations

The fulfill the goal of the study, several simulations have to performed with and without including the effects of both the urban
canopy meteorological forcing and the chemical fingerprint of the urban emissions (i.e. the UEI). Part of the simulations that
are analyzed in this work had been already analyzed in Huszar et al. (2020b): these included experiments with all emissions
considered but with or without considering the UCMF. As the focus of this paper is to evaluate the impact of urban emissions,
we extend these simulations by those without the inclusion of such emissions (for selected cities). The complete list of simula-
tions performed is included in Tab.1. The regional climate simulations included two experiments with the RegCM model, with
("URBAN") and without ("NOURBAN") considering the urban canopy meteorological effects (simulated by the CLMU urban
module), and, two experiments with the WRF-Chem model, again with and without considering urban canopies (simulated by
the SLUCM module).

The chemistry transport model simulations encompass CAMx runs based on RegCM and WRF-Chem regional climate
reconstructions and that corresponding to the "chemical" component of WRF-Chem. For each CTM, in total four simulations
are repeated. In two, the default emission data (i.e. all emissions) are considered while the inclusion of urban effect is once
included and than excluded. In other two, urban emissions for selected cities were completely removed (see Section. 2.2.2), i.e.
adopting the annihilation method (Baklanov et al., 2016). Finally, two additional simulations with CAMx driven by RegCM
meteorology were performed to analyze the effect of the UCMF induced changes of the vertical eddy diffusion (Kv) and their
contribution to the modelled urban emission impact.

This strategy of experimental design allow us to evaluate the UEI for both the URBAN and NOURBAN cases and analyze
their difference, or, in other words, to investigate how is the emission impact modulated by the UCMF. Thus, the main focus is





on the evaluation of:

$$\Delta UEI = UEI_{URBAN} - UEI_{NOURBAN}, \qquad (1)$$

where the UEI for URBAN or NOURBAN cases is evaluated for a pollutant $C$ as:

$$UEI_i = C_i(all\ emissions) - C_i(zero\ urban\ emissions), i \in \{URBAN,\ NOURBAN\}. \qquad (2)$$

The relative impact will be evaluated as

$$UEI_{i,rel} = \frac{C_i(all\ emissions) - C_i(zero\ urban\ emissions)}{C_i(all\ emissions)} \times 100\%, i \in \{URBAN,\ NOURBAN\}, \qquad (3)$$

for $NO_2$ and PM2.5, i.e. the relative contribution of urban emissions to the total concentration is provided. For $O_3$, $UEI_{i,rel}$ is calculated as

$$UEI_{i,rel} = \frac{C_i(all\ emissions) - C_i(zero\ urban\ emissions)}{C_i(zero\ urban\ emissions)} \times 100\%, i \in \{URBAN,\ NOURBAN\}, \qquad (4)$$

i.e. this number gives the relative change of the concentration after introducing urban emissions.

The relative $\Delta(UEI)$ presented throughout the manuscript is calculated as

$$\Delta(UEI)_{rel} = \frac{UEI_{URBAN} - UEI_{NOURBAN}}{UEI_{NOURBAN}}, \qquad (5)$$

### 2.2.2 Emission processing

For Europe, emissions provided by CAMS (Copernicus Atmosphere Monitoring Service) version CAMS-REG-APv1.1 inventory (Regional—Atmospheric Pollutants; (Granier et al., 2019)) for year 2015 were used as anthropogenic emission. For Czech republic, the high resolution national Register of Emissions and Air Pollution Sources (REZZO) dataset issued by the Czech Hydrometeorological Institute (www.chmi.cz) along with the ATEM Traffic Emissions dataset provided by ATEM (Ateliér ekologických modelů – Studio of ecological models; www.atem.cz) was used. These data offer activity based (SNAP – Selected Nomenclature for sources of Air Pollution) annual emission sums of main pollutants, namely oxides of nitrogen(NOx), volatile organic compounds (VOC), sulfur dioxide ($SO_2$), carbon monoxide (CO), PM2.5 and PM10 (particles with diameter less than 2.5 and 10 $\mu$m). CAMS are data defined on Cartesian grid. On the other hand, the Czech REZZO and ATEM datasets are defined as area, line (for road transportation) or point sources, while in case of the first these are usually irregular shapes that correspond to counties with resolution from a few 100 m to 1-2 km depending on the geometry of the particular shape, so appropriate for resolving urban emissions at 1 km domain resolution (in case of Prague).

Data from the listed emissions inventories are preprocessed using the FUME (Flexible Universal Processor for Modeling Emissions) emission model (http://fume-ep.org/; Benešová et al., 2018). FUME is designed primarily for preparation of CTM-ready emission files, including preprocessing the raw input files, the spatial redistribution of the data into the model grid, chemical speciation, and time disaggregation of input emissions. Category specific speciation factors (Passant, 2002) and time-dissaggregation (van der Gon et al., 2011) are applied to derive hourly speciated emissions for CAMx and WRF-Chem



models. Biogenic emissions of hydrocarbons (BVOC) for CAMx runs are calculated offline using the MEGANv2.1 (Model of Emissions of Gases and Aerosols from Nature version 2.1) emissions model (Guenther et al., 2012) based on RegCM and WRF meteorology. In case of WRF-Chem experiments, they are calculated online based on the MEGAN approach. The necessary
input for MEGAN (plant functional types, emission factors and leaf-area-index data) were derived based on Sindelarova et al. (2014).

To isolate the emissions originating from urban areas from those from elsewhere, various approaches are possible: one can select the gridboxes (or other shapes) in the source inventories that lie inside the city's limits or the same can be applied for the already redistributed emission on the model grid. Either approach is selected, first the city boundaries has to be properly defined.
For this purpose, we used the GADM public database (https://gadm.org) for the definition of administrative boundaries of the cities selected in this study. For the second task, the masking of inventory emissions based on the GADM shapes corresponding to cities, we had to ensure correct partition between the "city" and "non-city" portion of those shapes, which spawn over the city boundaries. For this purpose, the masking capability of FUME was used, which allows to defined arbitrary mask for subsetting emissions either inside or outside of the mask. To demonstrate the resulting masked emissions for the case of Berlin and Prague
at 9 km and 1 km resolution, we plotted in Fig. 2 the emissions of $NO_2$ for a selected hour. It is seen that (correctly) only those gridcells have zero emissions that are entirely encompassed within the city. Gridcells that have part falling off the city have non-zero emissions. City considered are Prague, Berlin, Munich, Budapest, Vienna and Warsaw.

## 3  Results

### 3.1  Model validation

As both the regional climate and CTM experiments presented here (those considering all emissions; see Tab. 1) were subject to validation in Huszar et al. (2020b), here we summarize only the most relevant conclusions. Regarding the meteorology, it was shown that each models performs reasonably within accepted range of biases comparable to other studies using very similar model configurations (Berg et al., 2013; Karlický et al., 2017, 2018). In general, RegCM precipitation is well captured in all seasons, while the winter temperatures are somehow larger than measured ones connected probably to reduced thermal cooling.
For WRF, precipitation is overestimated mainly during summer and is probably connected to overestimation riming caused by increased graupel sedimentation in deep convective clouds. Furthermore, both models show some overestimation of 10-m wind speeds and a reasonable reproduction of the maximum daily PBL heights with WRF performing better than the RegCM model which slightly overestimates it, caused probably by the strong vertical turbulent transport in the Holtslag scheme that RegCM uses.

Regarding the simulated air quality, in case of ozone ($O_3$), it is strongly overestimated given mainly by a nighttime positive bias (daytime values are captured reasonably) which emerges from inaccuracies in night-time chemistry and deficiencies in capturing nocturnal vertical eddy transport (Zanis et al., 2011). In general, WRF-Chem performs better in resembling the diurnal cycle of hourly ozone while both models encounter some negative bias in resolving extreme ozone values. $NO_2$ is systematically underestimated in all models suggesting either low emission values or incorrect $NO/NO_2$ speciation in the





emission model, supported also by the fact, that the diurnal cycles are well captured with respect to their shape (only the systematic underestimation persists mentioned earlier). It is also shown that high-resolution experiments are more successful in capturing the day-to-day variation of $NO_2$ values, probably as a result of higher resolution of emission data and also due to better representation of the terrain and hence the meteorological conditions. Finally, PM2.5 is usually underestimated in models and in both cold and warm seasons. This was attributed to underestimated nitrate aerosol formation and also to underestimated organic aerosol. As for the influence of these model deficiencies to the results, the underestimation of $NO_2$ and PM2.5 means that the UEI will be somewhat underestimated too in our models. In case of ozone, which is usually overestimated expect summer maximum values, it is expected that the average impact of UEI (decreases, see below) will be slightly underestimated in the model.

## 3.2 The impact of urban emissions

In our analysis we will focus on the near surface concentrations of $NO_2$, PM2.5 and $O_3$. First, the impact on seasonal – DJF (winter) and JJA (summer) averages will be presented. As different emissions and chemical regimes occur during the day, the diurnal cycles are of interest too. Further, given their high policy relevance, the impact on the extreme values will be evaluated as well. Finally, we will look how the vertical turbulent diffusion, as the most important component of UCMF, alone explains the modeled impact of (not-)considering the urban canopy meteorological effects.

### 3.2.1 Seasonal impact

In Fig. 3 the winter UEI for $NO_2$ is presented for the three applied modelling systems: RegCM/CAMx (9 and 1 km horizontal resolution), WRF/CAMx and WRF-Chem (both with only 9 km resolution). The first two columns shows the UEI evaluated from the "URBAN" experiments, the absolute impact shown in the first one while the relative contribution of urban emissions to the total concentrations in the second one. Higher values of the impact up to 4-6 ppbv are seen for each analyzed city and the similarities between models are very large. This corresponds to contribution of urban emission of around 40-60% for urban centers. The impact quickly becomes small further from cities with increases up to 1 ppbv over rural areas corresponding to about 5-10% contribution. The impact is not completely symmetric around cities owing to the prevailing wind directions. If calculating the UEI on the 1 km domain (case of Prague), it reveals some details of the emission structure of the city with maximum impact of 6 ppbv that correspond to 60-80% contribution so providing somehow larger relative impact for the city core resolved if higher resolution is applied. The UEI impact evaluated from the "NOURBAN" is evidently larger (3rd and 4th columns) exceeding 6 ppbv for each analyzed city and model too. In other words, the impact of city emissions is smaller if "URBAN" effects are considered and this difference can reach 2-3 ppbv, especially in the WRF-Chem experiments. The impact over areas surrounding cities is larger in the urbanized runs by about 0.1-0.2 ppbv, quickly becoming zero further from cities where the UCMF becomes negligible. In relative numbers, the decrease of UEI modelled for the URBAN case is about 20-40% and is seen for larger areas not confined to the analyzed cities. However, over areas where both the absolute city impact (UEI) and the difference $\Delta UEI$ is small, this relative decrease is a result of the ratio of two very small numbers and should be regarded with caution. This holds for areas where the relative change of $\Delta UEI$ is positive, which means stronger urban





emission impact for the "URBAN" compared to "NOURBAN" case. This is although explainable by the fact that in the former case, the stronger urban turbulence removes pollutants more effectively and the deposition occurs at larger distances leading
to concentration increase (seen in Huszar et al. (2020b)), although these changes are very small in absolute numbers so the relative difference between the "URBAN" and "NORUBAN" based UEI should be again treated with caution.

For the case of JJA in Fig. 4, the results are qualitatively very similar to DJF. The absolute impact of urban emissions is somewhat smaller reaching usually 4 ppbv being largest in the RegCM/CAMx model. The spatial extent of the impact larger than 0.05 ppbv is also smaller then in DJF. This is an expected consequence of in general smaller summer emissions due to
missing domestic heating source and also due to larger mixing into higher model levels allowing transport to distant areas. The relative contribution of urban emissions to the final concentrations is also smaller than in JJA. The UEI for the "NOURBAN" case leads again to higher emission impact reaching 6 ppbv (slightly higher then in the winter case). The spatial extent is however smaller then during DJF. This resulted in a different $\Delta UEI$ pattern in JJA: the spatial extent of the decrease for the "URBAN" case is smaller, however for urban cores, the decrease of the impact is larger compared to DJF. The relative change
of the UEI is also larger during this season often exceeding 60%.

For PM2.5 again, the general conclusions are similar to $NO_2$. The winter UEI reaches 4-6 $\mu gm^{-3}$ for the analyzed cities and reaches 0.5$\mu gm^{-3}$ for rural areas. The relative contribution of urban emission to the PM2.5 concentrations for the city centers and rural areas is about 20-40% and 1-5%, respectively. The UEI evaluated for the "NOURBAN" case is again stronger often exceeding 6$\mu gm^{-3}$, the rural impact is however very similar to the "URBAN" case. The decrease of UEI due to the inclusion
of urban effects (i.e. the UCMF) is usually between -2 and -3 $\mu gm^{-3}$ corresponding to about 40-60% smaller impact in the "URBAN" case compared to "NOURBAN" one for the city centers. The difference between the "URBAN" and "NOURBAN" UEI can be even positive, e.g. above rural Poland or also seen around Prague in the 1 km resolution runs. This can be explained by the fact that UCMF causes stronger turbulent removal of the urban emissions but this results in enhanced sedimentation further from cities leading to higher emission fingerprint there (especially in the WRF-Chem experiments).
During JJA, the PM2.5 urban emission induced concentration changes for both the "URBAN" and "NOURBAN" cases are resembling the situation in DJF quantitatively, but they are smaller, up to 2 $\mu gm^{-3}$ in all models and cities. They are larger reaching 4 $\mu gm^{-3}$ in the high resolution experiments for Prague where the concentrated character of sources is better resolved. In relative numbers, the contribution makes about 20-40% (except the city core in Prague at high resolution reaching 60%). If the UCMF is considered the UEI decreased about 1-2 $\mu gm^{-3}$ with exception of Prague in high resolution reaching -3 $\mu gm^{-3}$
decrease.

The impact of urban emission on regional ozone follows a different pattern than for $NO_2$ and PM2.5. As a secondary gas, it responses to increased emissions of its precursors (NOx and NMVOC) depend not only on the magnitude of each precursor but also on their ratio. In case CAMx (either driven by RegCM or WRF meteorology), introducing urban emissions resulted in clear $O_3$ decrease above the selected urban areas reaching -2 to -3 ppbv decrease as JJA average for the 9 km experiments,
while exceeding -4 ppbv for Prague at 1 km. In relative numbers, this represents a decrease by up to 10-20% (but usually between -5 and -10%) compared to the background concentrations (that does not consider urban emissions). The UEI over rural areas surrounding cities manifests itself as a slight increase of ozone by up to 0.5-1 ppbv (a few % in relative numbers).





For the "NOURBAN" case, the decrease of $O_3$ is stronger for each city usually exceeding -4 ppbv or even -6 ppbv in the high resolution runs for Prague. This means that the UEI for ozone is weaker in the "URBAN" case by around 2-3 ppbv.

For areas surrounding cities, where ozone increased due to urban emissions, the UEI difference between the "URBAN" and "NOURBAN" cases is positive too, meaning that the ozone increase due to UEI is larger in the "URBAN" case. In relative numbers, the change of UEI due to the introduction of the UCMF is often larger than -60% (stronger in the WRF-driven CAMx runs). For city vicinities, the relative change of UEI reaches high values too, up to 100% increase.

A different picture is gained if the UEI is evaluated for ozone based on the WRF-Chem model. The addition of urban
emission leads here to either no change or a little increase in the average summer ozone, indicating a different ozone isopleth pattern in case of the RADM2 mechanism used in the mentioned model (for more details, see the Discussion). For most of the analyzed cities, the UEI means about a 0.5 ppbv increase of ozone while for Berlin, it is rather characterized by a slight decrease (around -0.1 ppbv). These slight changes constitute a very small relative change of order up to 5%. The UEI is slightly smaller in magnitude in the "NOURBAN" case. In relative numbers one obtain a rather complicated pattern, however it is a result of
the ratio of very small, almost insignificant changes. What one can state for certainty that in the WRF-Chem model, urban emissions lead to rather slight ozone increase above cities which is stronger in the "URBAN" case than in the "NOURBAN" one.

### 3.2.2 Impact on diurnal cycles

As the UCMF has a different magnitude throughout the day (e.g. the urban warming or heat island has a clear peak during
early night hours while the urban boundary layer is thickest during the day), one might expect that the different of urban emission impact between the "URBAN" and "NOURBAN" cases will have a specific diurnal cycle too. We therefor calculated the average seasonal diurnal cycles of concentrations from the analyzed urban centres (averaged over all city).

Fig. 8 shows the DJF and JJA average diurnal cycles for surface $NO_2$ concentrations from individual simulations and their differences (i.e. the UEIs and the $\Delta$UEIs). In general, the three models perform in a very similar manner. The UEI impacts
for both the "NOURBAN" (blue) and "URBAN" (orange) cases follow strongly the typical pattern for the NOx emissions exhibiting two peaks during morning and afternoon rush hours (the weekends have somewhat weaker pattern but weekdays dominate the average). It is clear that the UEI for "URBAN" case (orange dashed) is much lower reaching 10-15 and 5-10 ppbv as peaks in DJF and JJA, respectively while the difference with respect to the "NOURBAN" based UEI (blue dashed) is varying during the day. Its exact diurnal pattern is shown by the green line which has a very clear pattern in both seasons
and all models (belongs to right vertical axes). In absolute sense its almost zero during night-time and reaches its maximum during early evening hours. In RegCM driven CAMx the maximum reaches -3 to -4 ppbv for DJF and JJA, respectively while much stronger decrease is modeled with WRF meteorology reaching -12 ppbv. This is much greater change of UEI (due to the UCMF) than seen for the seasonal means in Fig. 3 and4. A second, smaller peak during morning hours is also exhibited by each models that reaches values between -1 to -4 ppbv.

A qualitatively very similar result is obtained for PM2.5, Fig. 9. Again, the three models provide comparable results. The UEI impacts for both the "NOURBAN" and "URBAN" cases follow strongly the typical pattern for the PM emissions ex-





hibiting two peaks during rush hours. It is clear that the UEIs for "URBAN" case are much lower reaching 4-6 and 2-3 ppbv as peaks in DJF and JJA, respectively, while the difference with respect to the "NOURBAN" UEI case is varying during the day. Its exact diurnal pattern is shown by the green line which, again, has a very clear pattern in both seasons and all models.

In absolute sense its almost zero during night-time and reaches its maximum during early evening hours. In RegCM driven CAMx the peak reaches -1.4 and -1.2 $\mu gm^{-3}$ for DJF and JJA, respectively while much stronger decrease is modeled with WRF meteorology reaching -5 $\mu gm^{-3}$ in DJF (-3.5 $\mu gm^{-3}$ during JJA). Similarly to $NO_2$, this is a much greater change of UEI (due to the UCMF) than seen for the seasonal means in Fig. 5 and 6. A second, usually much smaller peak during morning hours is also exhibited by each model that reaches values between -1 to -2 $\mu gm^{-3}$.

In case of ozone, for the CAMx experiments, the UEI shows a strong correlation with the absolute values (dashed vs. solid lines) meaning that when urban ozone values are lowest, also the UEI for $O_3$ shows its maxima (in absolute sense). The maximum impact occurs during morning and early evening hours reaching -12 ppbv and -10 ppbv for DJF for the "NOURBAN" and "URBAN" cases, respectively, and reaching minimum values during noon and night. During JJA, the UEI reaches -10 pbv and -8 ppbv for the "NOURBAN" and "URBAN" cases while the UEI for the "URBAN" case is smaller in the WRF driven

CAMx experiment (reaching -5 ppbv). In conclusion, in CAMx driven either by RegCM or WRF, the UEI is negative during the whole day. In case of WRF-Chem, the impact is expectedly negative throughout the whole day in DJF, but for summer, it becomes positive during daytime reaching 1.5 ppbv (the "URBAN" case being slightly higher). During night, the UEI reaches negative values up to -1 ppbv for the "NOURBAN" case and -0.2 ppbv for the "URBAN" one. The $\Delta$UEI (green line) is rather positive in each model and season showing a clear maximum during late afternoon/evening hours, which reaches a few ppbv in

DJF (0.5 in WRF-Chem) while in JJA, the maximum can be as high as 10 ppbv. This indicates that the urban emission impact for ozone is smaller in absolute sense if the model predicts its decrease due to UEI (CAMx), while it is higher if the model predicts increase (WRF-Chem).

### 3.2.3   Impact on extreme values

Huszar et al. (2020b) showed that the urban canopy meteorological forcing has a stronger effect on extreme air pollution (95th

percentile of $NO_2$, PM2.5 and $O_3$) compared to the average one. This motivates us to look also at the UEI for such situations. We therefore plotted the UEI on the 95th percentile values of the daily means of the analyzed pollutants and were interested also in the associated differences between the UEI evaluated for the "NOURBAN" and "URBAN" cases (i.e. $\Delta$UEI). In Fig. 11 we see that the UEI for the 95th percentiles of $NO_2$ is much higher compared to the impact on average one and reaches 9-12 ppbv, especially in the WRF-Chem model (left column). It can be also seen that it reaches 4-6 ppbv (i.e. the values seen for the

averages) at distances roughly twice of the city size indicating crucial rule urban emission play at extreme air pollution events over regional scales. Regarding the modulation of the emission impact by the UCMF (the right column), it is seen that it is also much higher compared to the difference in case of averages and can reach -6 ppbv. This means that the UEI on extreme air pollution is even more strongly reduced by the urban effects that the averages seen in Fig. 3-4.

    In case of PM2.5 in Fig. 12, the UEI for 95th percentiles is again higher than calculated for the averages and reaches 8

$\mu gm^{-3}$ in each model (especially for Prague, Budapest and Warsaw). This is again higher than the impact on averages (up to





6 μgm$^{-3}$) and points to increased role of urban emission during extreme PM pollution events. Our results also suggest a large impact over rural areas reaching 1-2 μgm$^{-3}$ indicating potential for urban emissions to enhance rural concentrations too. The decrease of the UEI on extreme values of PM2.5 if the UCMF is considered, can be as large as -4 to -6 μgm$^{-3}$. Again, this is a stronger decrease compared to the values obtained for the DJF and JJA averages (Fig. 5-6).

In case of ozone, Fig. 13, somehow different behavior is modeled for CAMx and for WRF-Chem (similarly to the impact on seasonal averages) but differences are encountered between cities too. The UEI for 95th percentile of the daily maximum 8-hour O$_3$ is usually negative over cities reaching -4 ppbv over city cores – this is the case of RegCM/CAMx and partly in WRF/CAMx too (for Berlin, Warsaw, Prague and Budapest) with smaller decrease up to -1 ppbv. These are smaller decreases that those seen for average JJA ozone in Fig/ 7. On the other hand, for cities like Budapest and Berlin in WRF/CAMx experiments and for

all cities in WRF-Chem, the UEI for 95th ozone values is positive with increases up to 2-4 ppbv in WRF-Chem. For areas surrounding cities, further from the origin of the emissions, there is a clear increase of ozone reaching 1 ppbv over large areas. This indicates that during extreme ozone periods, urban emissions push the balance between the ozone production and reduction towards production leading to smaller reductions for some cities and models and stronger increase for other cities and models. The change between the "URBAN" and "NOURBAN" cases is positive for RegCM/CAMx similarly to the impact

on averages, meaning that the reduction of extreme ozone is smaller when the UCMF is considered but this modification is smaller than seen for average values. For other models, the modulation of UEI between "URBAN" and "NOURBAN" is rather small and can be negative or positive with the preference of negative change, i.e. enhancement of the UEI in case of the WRF-Chem model. This means that if extreme ozone responds to urban emissions with increase, the increase is smaller if UCMF are considered. On the other hand, if ozone responds with decrease, then this decrease is smaller in absolute values when

non-considering UCMF.

### 3.2.4    The role of vertical turbulence

As seen in previous sections, the urban emission impact is significantly perturbed if the urban canopy meteorological forcing is considered in its model estimation and is usually overestimated if UCMF is disregarded. Many previous studies showed that the most important component of the UCMF influencing the urban air pollution is the vertical eddy-diffusion (e.g. Zhu

et al., 2015; Huszar et al., 2020a). To evaluate its role and contribution to the changes of UEI between the "URBAN" and "NOURBAN" cases, we performed additional simulations with the effect of perturbed vertical eddy-diffusion coefficients (Kv) removed (denoted as "NKV"). To estimate the effect of Kv changes alone, we computed the UEI for the "NKV" case too and compared it to UEI obtained for the "URBAN" case.

     In Fig. 14, the diurnal variations of the UEIs calculated for the "NOURBAN" and "URBAN" cases as well as for the "NKV"

case. Besides, it shows the modulation of the UEI due to the consideration of all components of the UCMF as well as due to consideration of only of the Kv effects. In case of NO$_2$, it is evident that the diurnal cycle of the UEI calculated for the "NOURBAN" case is very close the the cycle for the "NKV" case, and this holds for both DJF and JJA. In other words, the modulation of this impact by considering all urban effects ("URBAN" case) is almost the same if taking for reference the "NOURBAN" or "NKV" cases. This means that the vertical eddy-diffusion component of the UCMF alone explains the





modeled modulation of UEI due to the UCMF. The differences arise from the fact, the UCMF also contains the component of decreased wind and increased temperature (Huszar et al., 2018b, 2020a) and especially the wind's effect causes that the urban concentrations are higher due to lower windspeeds and limited dispersion. This implies a higher UEI in the "NKV" case compared to "NOURBAN" case because wind stilling is present. Adding the Kv effects leads to slightly lower UEI but it is clear, that the turbulent eddy-diffusion dominates the UEI.

The situation is similar for PM2.5, showing that the diurnal cycle of UEI for the "NOURBAN" case is very close to the cycle calculated for the "NKV" case with highest differences during the morning and afternoon peak hours. Consequently, the modulation of UEI due to all components of UCMF vs. due to Kv effects only is very similar, especially during JJA. Finally, the situation for $O_3$ follows the conclusions for the previous two pollutants. The UEI based on the "NOURBAN" and "NKV" shows very similar decreases with differences less then 0.5 ppbv. This implies that the $\Delta$UEI due to all urban effects is again

very close to that due to Kv effects only, especially during JJA.

In summary, the modulation of the urban emission impact due to the UCMF is largely determined by the acting of the enhanced vertical eddy-diffusion. The effect of other components is small and explain the slight difference between the UEI calculated for the "NOURBAN" vs. "NKV" cases.

## 4 Discussion and conclusions

This study looked at the regional air-quality impact of urban emissions (UEI) from selected large cities and agglomerations in central Europe with the focus of not only quantifying the UEI magnitude but also the modulation of UEI due to the consideration of the urban canopy meteorological forcing (UCMF). The UCMF was calculated from a pair of runs (with and without considering urban land-cover) for two regional climate models, RegCM and WRF-Chem, while the UEI was quantified using CAMx driven by both of these models and also by WRF-Chem itself, using the annihilation method meaning that urban

emissions were completely removed in the reference run and compared to the full emission run.

Before discussing the obtained results, the model's performance with respect to measurements has to be evaluated. This study did not explicitly performed a model evaluation because the "full" emission model runs were already validated in Huszar et al. (2020b). Based on comparison to urban ground sites, they found a systematic underestimation of both $NO_2$ and PM2.5 throughout the year and also in the daily cycles. This indicates underestimation of urban emissions and consequently it also

means the UEI simulated in this study might be somehow underestimated. Similar underestimation of PM2.5 was also encountered in Ďoubalová et al. (2020) who applied CAMx coupled offline to WRF using almost identical emissions over roughly the same domain, and previously by Huszar et al. (2018a, b) and Huszar et al. (2020a) too. The modeled negative PM2.5 bias can be attributed to underestimation aerosol components like nitrates and organic aerosol, as seen - similarly to our results - in Schaap et al. (2004); Myhre et al. (2006). However, besides underestimated emission inventory values, too strong daytime

dilution and overestimated vertical turbulence can probably play role too as argued by (Nopmongcol et al., 2012). Indeed, there is a large uncertainty in calculating the vertical eddy-diffusion for urban pollutants (Huszar et al., 2020a) which can strongly reduce near surface concentrations if being too strong. Regarding $NO_2$, the underestimation is very similar to Karlický et al.





(2017) and Tucella et al. (2012) who used WRF-Chem over Europe too, although with slightly different emission inventory data.

The validation in Huszar et al. (2020b) also showed strong overestimation of ozone in monthly means caused mainly by the overestimation of night-time ozone while daytime values are captured reasonably. This behavior was attributed to deficiencies in nighttime chemistry and also inaccurately resolved vertical mixing in the nocturnal boundary layer (Zanis et al., 2011). On the other hand, the maximum 8-hour ozone values were underestimated in Huszar et al. (2020b) which points to the fact that models are unable to resolve the highest ozone values correctly. This is also one of the main conclusions in the comprehensive

AQMEII model inter-comparison presented by Im et al. (2015). As nighttime ozone is overestimated in our simulations, we can expect that the titration effects are underestimated. Consequently, the simulated $O_3$ decreases due to UEI are also probably underestimated.

Our results showed that the urban contribution of $NO_2$ can reach 40-60% in urban cores meaning that roughly half of the urban $NO_2$ originates from elsewhere (rural sources, smaller cities etc.). This contribution corresponds to numbers in Huszar

et al. (2016a) who calculated an average annual impact of 42% averaged over a large number of European cities. Im and Kanakidou (2012) modelled even higher contribution: 90-95% for eastern Mediterranean cities indicating that the role of local emission can be even higher, especially in cities with higher rural-urban contrast, or if higher resolution is used. Indeed, our 1 km runs for Prague showed 60-80% contribution. As the emitted NOx quickly decay to $HNO_3$ as the urban plume is diluted, the contribution becomes very small further from cities reaching around 5-10%, again, very similar to values in Huszar et al.

(2016a) or Guttikunda et al. (2005). Although not analyzed in this study, but Guttikunda et al. (2003) found similar values of contribution for $SO_2$.

More importantly, we showed that by considering the urban canopy meteorological forcing, the contribution of urban $NO_2$ becomes lower by about 20-40%. This is an important result as it clearly says that the impact of urban emission is overestimated if the urban canopy is not properly represented in regional modelling. This further means that the background concentrations

(i.e. those with zero urban emissions) are not so strongly affected by UCMF than the "full" emissions, generating this modification of the UEI. This goes in line with previous studies that looked at the UCMF impact on urban $NO_2$ levels. E.g. Huszar et al. (2018a) gave decreases by about 1-2 ppbv similarly to our change of the UEI. Struzewska and Kaminski (2012) also found a comparable reduction of $NO_2$ due to UCMF, as well as Kim et al. (2015) who showed that due to increased turbulence over cities, $NO_2$ levels are strongly reduced.

The general conclusion for PM2.5 are very similar to $NO_2$. The contribution of urban emissions in the analyzed cities is around 20-40%, which is similar than the values in Huszar et al. (2016a) who simulated 30-60% contribution. Again, larger contribution, in both relative and absolute numbers was simulated by Im and Kanakidou (2012), probably due to lower background pollution. The contribution to rural areas is also similar to the mentioned studies (around 1-5%). Our analysis showed, analogously to $NO_2$, that if the urban canopy effects are considered, the impact on PM is smaller by around 50%.

In other words, if the UCMF is not considered, the impact of urban emissions on PM is strongly overestimated. The reasons for this is again in the acting of urban canopy by modifying vertical eddy diffusion, wind-speeds, temperature. Indeed, Huszar et al. (2018b, 2020a); Kim et al. (2015); Zhu et al. (2017); Wei et al. (2018) all modelled decreased aerosol concentration





over urban areas as a result of considering the urban canopy effects and further identified, that the most important factor that contributes to this decrease is the enhanced vertical turbulence.

In case of the impact on $O_3$, different behavior is encountered between the models. While CAMx responds to decreases of near surface concentrations, a distinct behavior is seen for WRF-Chem with almost no change in concentrations or a slight increase of average JJA ozone. As ozone is a secondary pollutant, it's response to elevated emissions (like adding urban emissions to rural ones) can have both signs depending on the ratio of the precursor emissions: NOx and NMVOC. Cities are characterized with a high NOx−to−VOC ratio (Beekmann and Vautard, 2010) and the response to emission changes depend

on the slope of ozone isopleths at a given initial NOx/NMVOC ratio. CAMx applied the Carbon Bond 5 chemical mechanism and Nopmongcol et al. (2012) showed that 100% increase of emissions leads to ozone decreases in their analysis based on ozone isopleth for London and summer conditions. This is in line with our study also leading to ozone decreases using the same chemical mechanism. Also Huszar et al. (2016a) for a large number of central European cities and Im et al. (2011a, b) for Mediterranean cities showed similar decreases of ozone in urban cores. As already said, in case of WRF-Chem ozone

responded in a different way. For RADM2 mechanism, which is used by WRF-Chem, Stockwell et al. (2011) showed based again on ozone isopleth analysis that for emissions changes of NOx and NMVOC at ratio NOx/NMVOC around 0.2 the ozone change is minimal. Indeed, for our cities, the average NOx/NMVOC emission ratio is between 0.15-0.25. This explains the minimal ozone response modelled for WRF-Chem above cities. For both CAMx and WRF-Chem, further from cities, urban emission lead to ozone increases. This is not surprising as when urban NOx-rich emissions are diluted and mixed with rural

emission with higher NMVOC/NOx, ozone production occurs (Poupkou et al., 2008).

    Regarding the modulation of the urban emission impact on $O_3$ due to UCMF, the decrease seen in CAMx became smaller, almost by 60%. In other words, the ozone reduction in city core is simulated too high if urban canopy effect are not considered in the regional climate model calculations. For WRF-Chem, as the ozone response was weak, the consideration of urban effects lead to only a slight increase of the impact, i.e. if the UCMF is not considered, the ozone response to urban emissions is smaller.

In either CAMx or WRF-Chem, the ozone response to the inclusion of UCMF is in line with previous studies dealing with the urban canopy effects in air-quality. Wang et al. (2007, 2009); Ryu et al. (2013); Huszar et al. (2018a); Li et al. (2019) simulated ozone increase due to UCMF too and attributed it mainly to the dominating effects of increased vertical removal of NOx and consequent reduced titration while the nocturnal downward turbulent flux of residual ozone can be also important (Huszar et al., 2018a).

The diurnal cycles revealed and important fact: the decrease of the UEI due to the UCMF is not uniform during the day but has a clear maximum during late afternoon/evening hours for both $NO_2$ and PM2.5. This caused by the timing of the maximum effect of UCMF seen in Huszar et al. (2018a, b). As shown in Kim et al. (2015) and more in detail in Huszar et al. (2020a), the modifications of the vertical eddy-diffusion coefficients, which is the dominant component of the UCMF, are largest during these times of day, counteracting the decrease of wind-speed during evening hours (Karlický et al., 2018).

This means that the largest vertical mixing of these pollutants occur during these times and hence the largest reduction of the emission impact occurs. A secondary maximum of the UEI reduction is encountered during morning hours. During that the turbulence enhancement is rather small but the emissions are high due to the morning rush hour and hence the high modification



of their impact which is proportional to the quantity of the emitted pollutants. In case of ozone, the maximum of the UEI change also occurs during late afternoon/evening hours and it is again most probably related to the maximum of the modifications of

vertical turbulence.

Indeed, our sensitivity analysis focusing on the isolated impact of vertical eddy diffusion modifications showed very similar diurnal cycles for the UEI modulation than those caused by the total UCMF impact (which includes additional effects of temperature, wind, moisture etc., see (Karlický et al., 2020)). This confirms the conclusion of studies about the role of urban turbulence triggered either mechanically or thermally (via urban heat island, anthropogenic heat etc.), which strongly modulates

their air pollution (Kim et al., 2015; Zhu et al., 2015, 2017; Xie et al., 2016a, b).

Based on this study, we can conclude that the local and regional fingerprint of urban emissions over central Europe is strong but large differences arise from whether the urban canopy related meteorological effects, like enhanced turbulence, urban heat island, reduced wind-speed etc., are considered. Based on three modeling systems, we showed that the impact on near surface $NO_2$ and PM2.5 is reduced if they are taken into account. In case of ozone, the the modulation of the emission impact depends

on the emission impact itself: if it is negative and ozone decreases due to urban emissions, than this decrease is reduced if UCMF is considered. On the other hand, if ozone increases due to urban emissions, UCMF causes this increase to be even larger. In any case, we argued that the urban canopy and all the resulting effects on meteorological processes have to be properly accounted for in regional models when the transport of pollutants from urban areas is studied and the impact of such emissions is quantified.

*Code and data availability.* The RegCM4.7 model is freely available for public use at https://gforge.ictp.it/gf/download/frsrelease/259/1845/RegCM-4.7.0.tar.gz (Giuliani, 2021). CAMx version 6.50 is available at http://www.camx.com/download/default.aspx (ENVIRON, 2018). The wrf-fcamx preprocessor is available from http://www.camx.com/download/support-software.aspx. WRF-Chem version 4.1 can be downloaded from https://www.acom.ucar.edu/wrf-chem/download.shtml (WRF, 2020). The RegCM2CAMx meteorological preprocessor used to convert RegCM outputs to CAMx inputs is available upon request from the main author. The complete model configuration and all the simulated

data (3-dimensional hourly data) used for the analysis are stored at the Dept. of Atmospheric Physics of the Charles University data storage facilities (about 40TB) and are available upon request from the main author.

*Author contributions.* PH provided the scientific idea, the design of the model experiments, the project coordination, and supervised the writing of the paper; PH, JK were responsible for performing the RegCM, CAMx and WRF-Chem experiments; JM, TN, LB and ML contributed to the evaluation of the results.

*Competing interests.* The authors declare that they have no conflict of interest.



*Acknowledgements.* This work has been funded by the Czech Science Foundation (GACR) project No. 19-10747Y and partly by projects PROGRES Q47 and SVV 260581/2020 – Programmes of Charles University. We further acknowledge the CAMS-REG-APv1.1 emissions dataset provided by the Copernicus Monitoring Service, the Air Pollution Sources Register (REZZO) dataset provided by the Czech Hydrometeorological Institute and the ATEM Traffic Emissions dataset provided by ATEM (Studio of ecological models). We acknowledge the E-OBS dataset from the EU-FP6 project UERRA (http://www.uerra.eu) and the Copernicus Climate Change Service, the data providers in the ECA&D project (https://www.ecad.eu) and the data providers of AirBase European Air Quality data (http://www.eea.europa.eu/data-and-maps/data/aqereporting-1) and AIM (Automatic Imission Monitoring network data http://portal.chmi.cz/aktualni-situace/stav-ovzdusi/prehled-stavu-ovzdusi?l=en)



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





**Table 1.** The list of model simulations performed: the first section contains the RCM simulations that cover the whole analyzed period with the information whether urban land-surface was considered (2nd column). The second section lists the performed regional CTM experiments – here the second column provides information on the driving meteorological data (not needed in case of WRF-Chem).

| Regional Climate Model (RCM) runs | | |
|---|---|---|
| Model | Urbanization[a] | Resolution[km] |
| RegCM | YES | 9/3/1[b] |
| RegCM | NO | 9/3/1 |
| WRF-Chem | YES | 9 |
| WRF-Chem | NO | 9 |

| Regional Chemistry Transport Model (CTM) runs | | | |
|---|---|---|---|
| Model | Emission scenario | Driving Data | Resolution[km] |
| CAMx | all | RegCM9U(/3U/1U) | 9/3/1 |
| CAMx | no urban | RegCM9U(/3U/1U) | 9/3/1 |
| CAMx | all | RegCM9NU(/3NU/1U) | 9/3/1 |
| CAMx | no urban | RegCM9NU(/3NU/1U) | 9/3/1 |
| WRF-Chem URBAN | all | –[c] | 9 |
| WRF-Chem URBAN | no urban | – | 9 |
| WRF-Chem NOURBAN | all | – | 9 |
| WRF-Chem NOURBAN | no urban | – | 9 |
| CAMx | all | WRF-Chem URBAN | 9 |
| CAMx | no urban | WRF-Chem URBAN | 9 |
| CAMx | all | WRF-Chem NOURBAN | 9 |
| CAMx | no urban | WRF-Chem NOURBAN | 9 |
| CAMx | all | RegCM9NKV[d](/3NKV/1NKV) | 9/3/1 |
| CAMx | no urban | RegCM9NKV(/3NKV/1NKV) | 9/3/1 |

[a]Information whether urban land-surface was considered.
[b]Simulation performed in a nested way on 9, 3 and 1 km.
[c]No driving meteorological data needed as chemistry is online coupled to the parent meteorological model
[d]NKV - not considering the urban induced modifications of the vertical eddy-diffusion coefficients





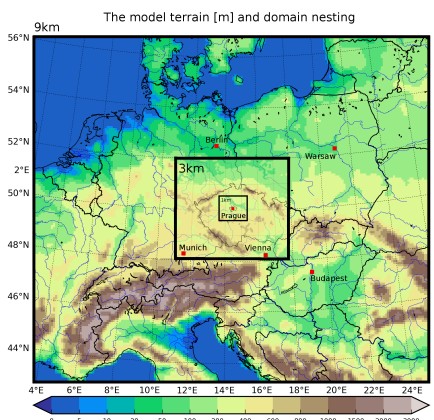

**Figure 1.** The resolved model terrain in meters, the nesting structure and the cities analyzed in the study (Prague, Berlin, Munich, Vienna, Budapest and Warsaw).

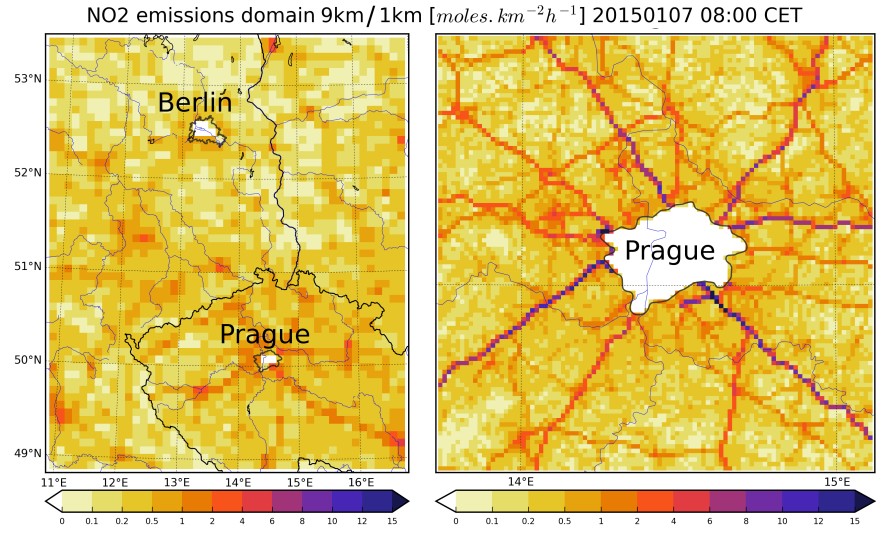

**Figure 2.** Demonstration of the masked $NO_2$ emissions for Prague and Berlin for the 9 km and 1 km domains in $mol\,km^{-2}\,h^{-1}$ (shown only a part of the domains).



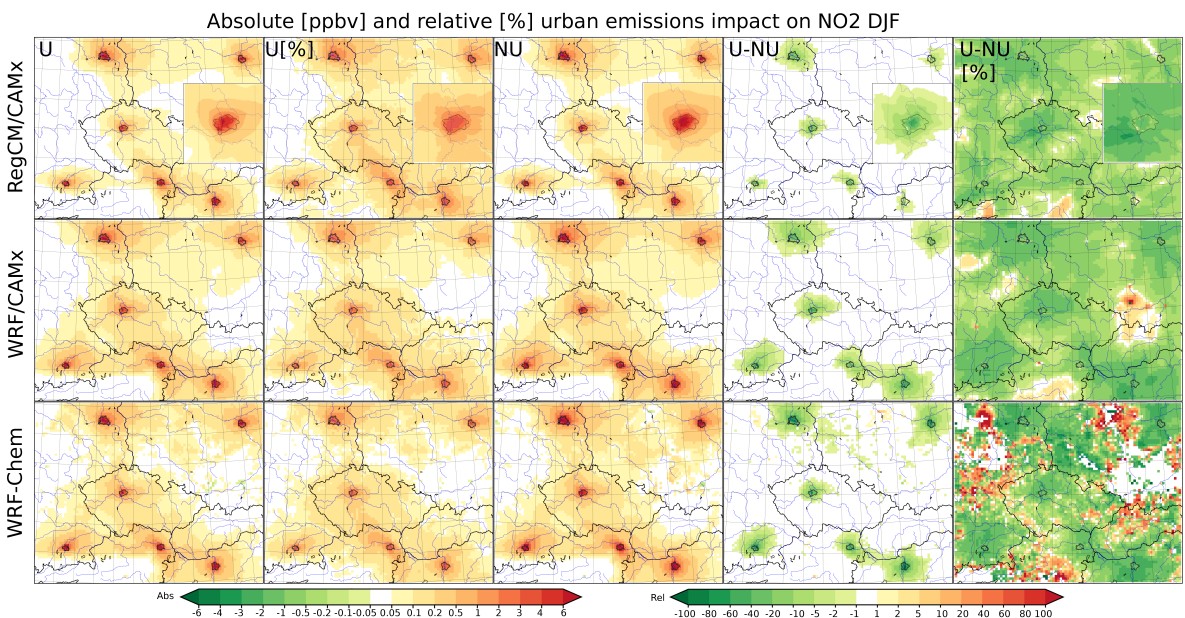

**Figure 3.** The urban emission impact (UEI) of five selected cities (Berlin, Budapest, Munich, Prague, Vienna and Warsaw) on 2015-2016 average DJF near surface $NO_2$ concentrations for the three modelling systems used: CAMx driven by RegCM (1st row), CAMx driven by WRF-Chem meteorology (2nd row) and WRF-Chem (3rd row). Individual columns represent: the UEI evaluated for the "URBAN" experiments as absolute ($UEI_{URBAN}$; U; in ppbv) and relative impact ($UEI_{URBAN,rel}$; U[%]), the UEI impact for the non-urbanized ("NOURBAN") experiments ($UEI_{NOURBAN}$; NU), the difference between the the two ($\Delta UEI$; U-NU) and the relative change of the impact ($\Delta(UEI)_{rel}$) in %. The corresponding results for Prague at 1 km resolution from the RegCM driven CAMx experiments are plotted within the 9 km figures in the upper row.



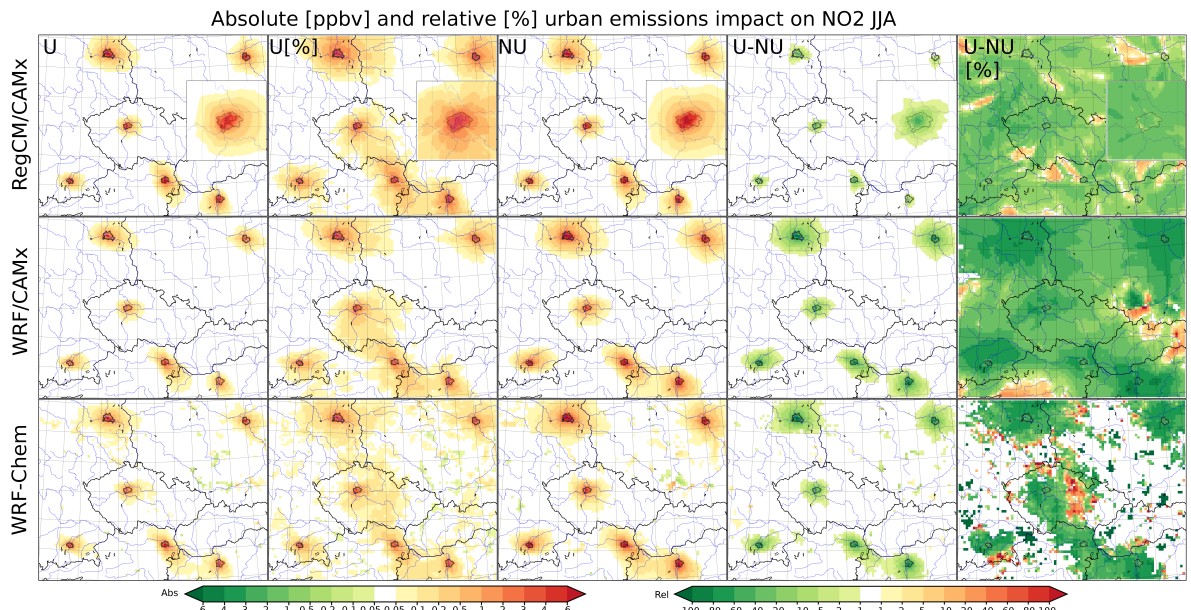

**Figure 4.** Same as Fig. 3 but for JJA.

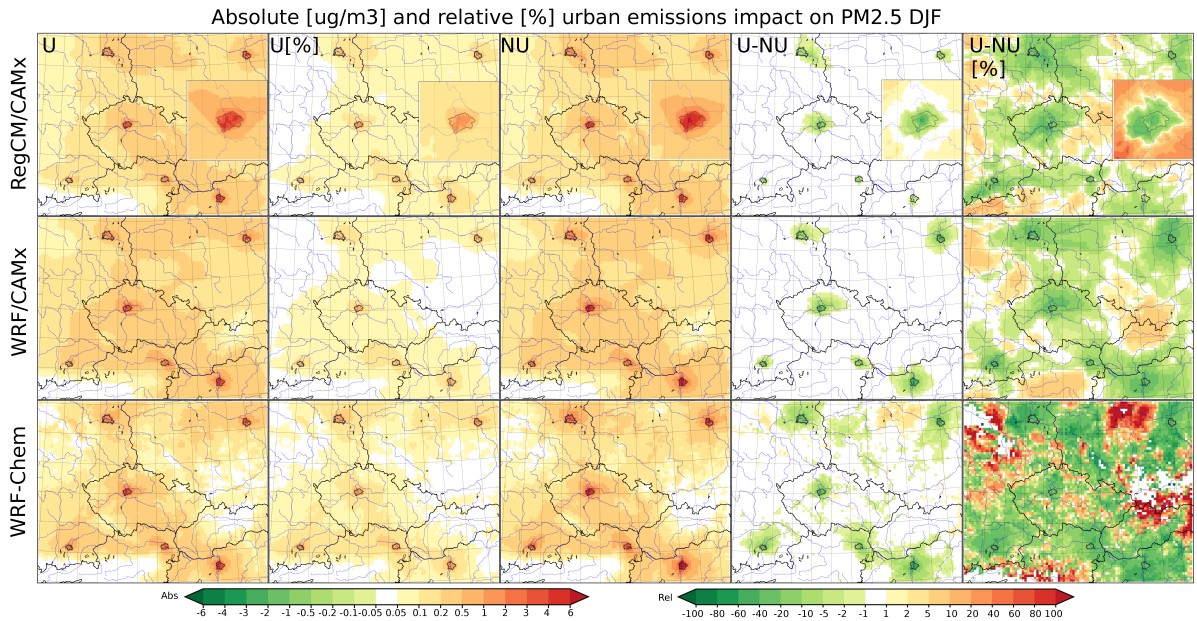

**Figure 5.** Same as Fig. 3 but for PM2.5 and in $\mu gm^{-3}$.





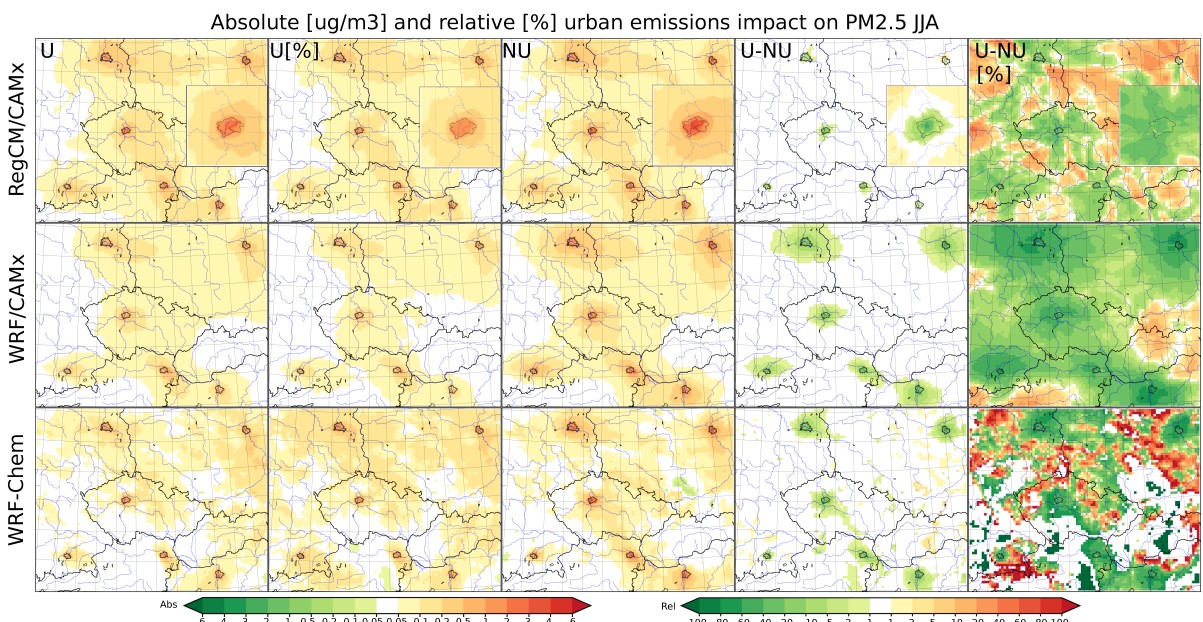

**Figure 6.** Same as Fig. 5 but for JJA.

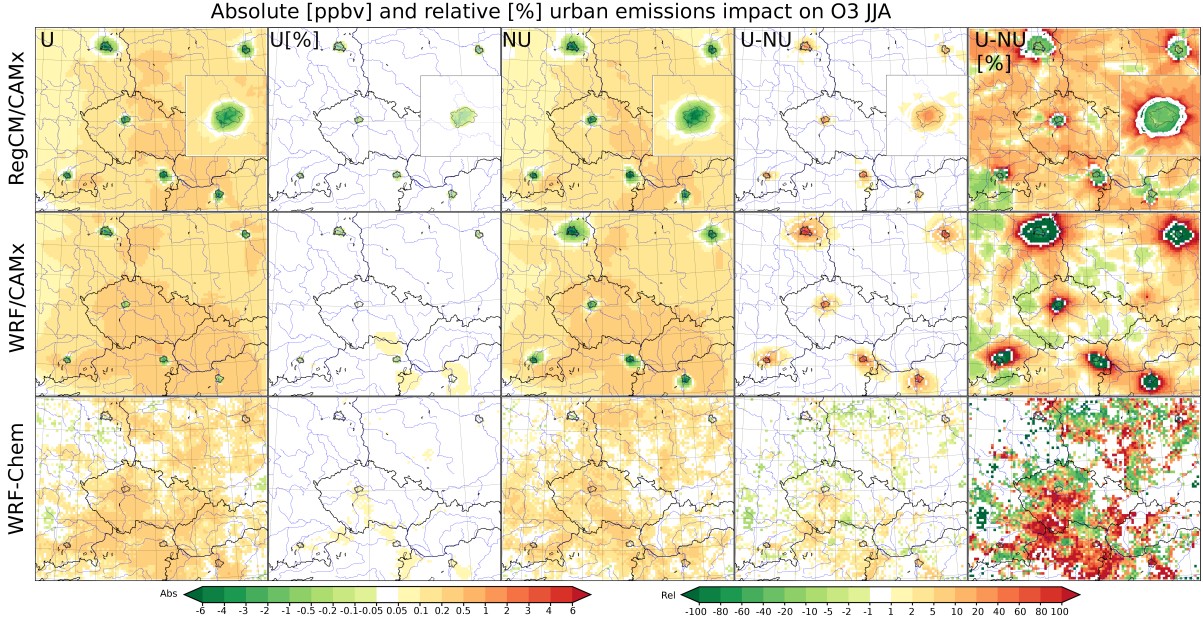

**Figure 7.** Same as Fig. 4 but for $O_3$.



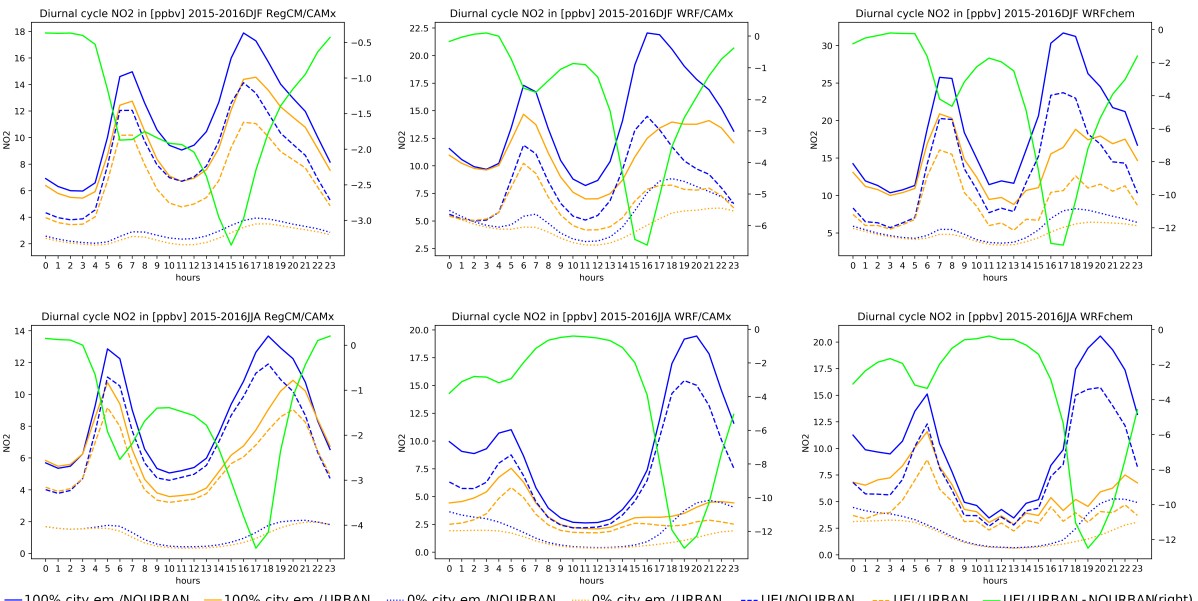

**Figure 8.** The average diurnal cycle of the absolute urban $NO_2$ concentrations and their difference between different simulations for DJF (upper row) and JJA (lower row) and for the three models applied as columns: CAMx driven by RegCM, CAMx driven by WRF and WRF-Chem. Blue/orange lines denote the "NOURBAN/URBAN" case. Solid lines stand for the reference values with 100% urban emissions, dotted lines for the 0% urban emission runs and dashed ones for the UEI evaluated for both the "NOURBAN" and "URBAN" case. These correspond to the left vertical axis. Finally, the green line denotes the $\Delta$UEI (right vertical axis), i.e. the modification of the urban emission impact due to the inclusion of the urban effects (the UCMF). Units in ppbv.





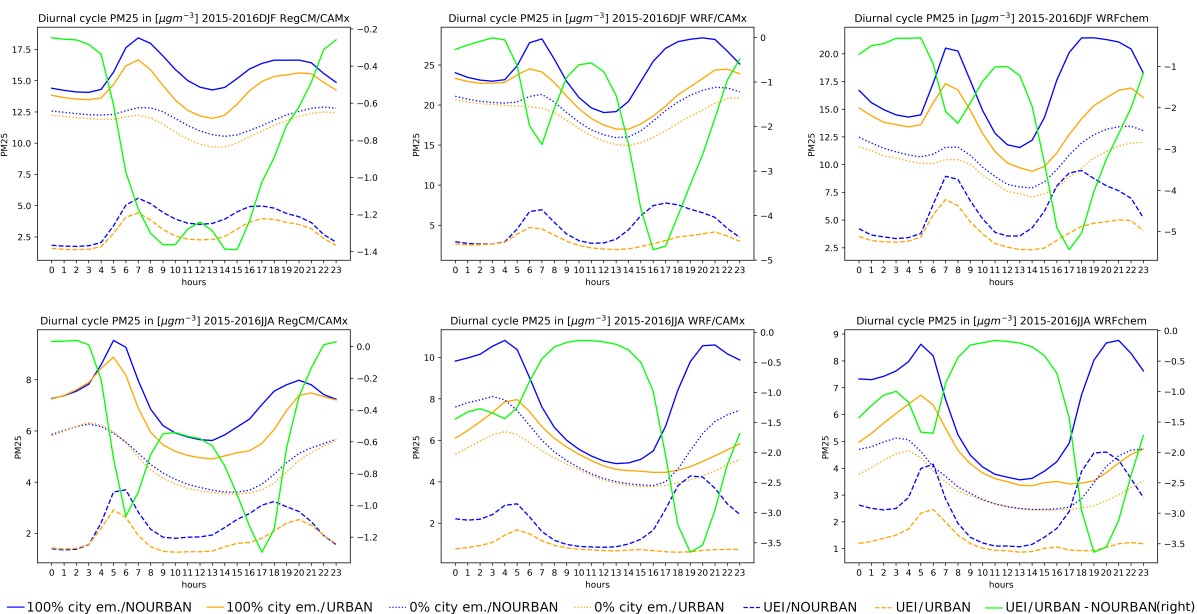

**Figure 9.** Same as Fig. 8 but for PM2.5 and in $\mu gm^{-3}$.



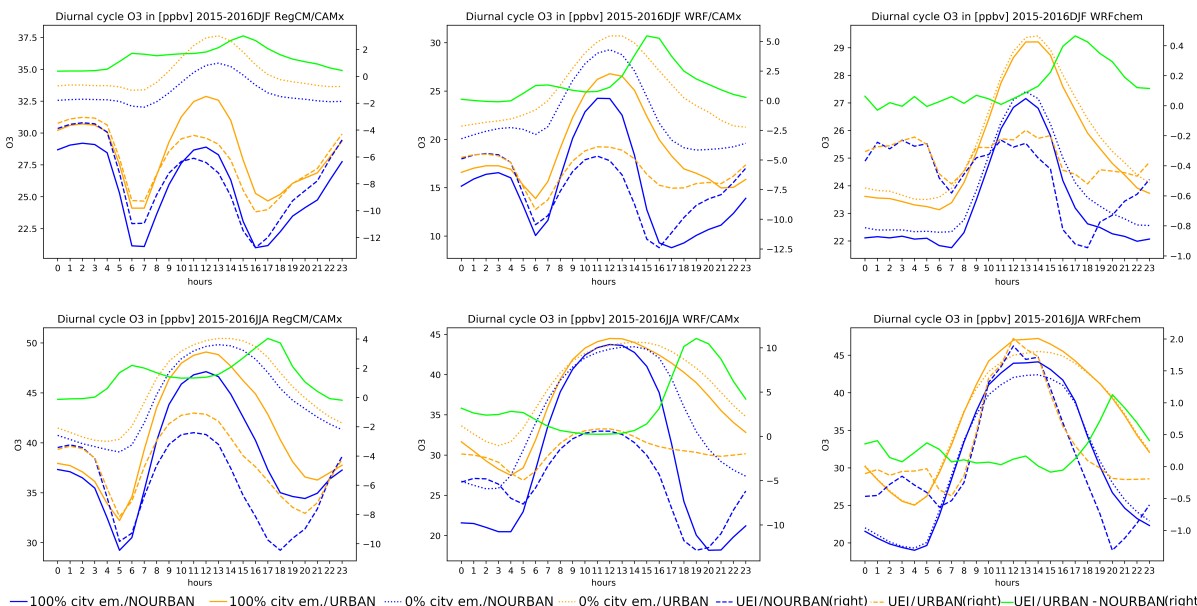

**Figure 10.** The average diurnal cycle of the absolute urban O$_3$ concentrations and their difference between different simulations for DJF (upper row) and JJA (lower row) and for the three models applied as columns: CAMx driven by RegCM, CAMx driven by WRF and WRF-Chem. Blue/orange lines denote the "NOURBAN" and "URBAN" case. Solid lines stand for the reference values with 100% urban emissions while dotted lines for the 0% urban emission runs (both belonging to the left vertical axis). Dashed lines mean the UEI evaluated for both the "NOURBAN" and "URBAN" case and the green line denotes the $\Delta$UEI (right vertical axis). Units in ppbv.

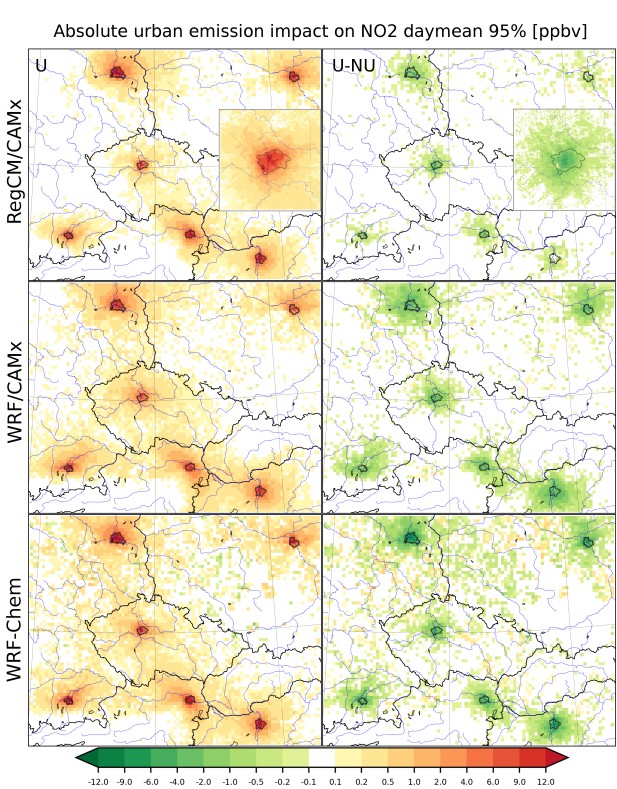

**Figure 11.** The urban emission impact (UEI) of selected cities (Berlin, Budapest, Munich, Prague, Vienna and Warsaw) on the 95% percentile of the daily mean $NO_2$ concentrations for the three models (as rows) for the 2015-2016 period: the left columns shows the absolute UEI for the "URBAN" case while the right columns denotes the $\Delta$UEI, i.e. the difference between the UEI for the "URBAN" and "NOURBAN" cases. Results from the 1 km x 1 km experiment is plotted within the upper row. Units in ppbv.

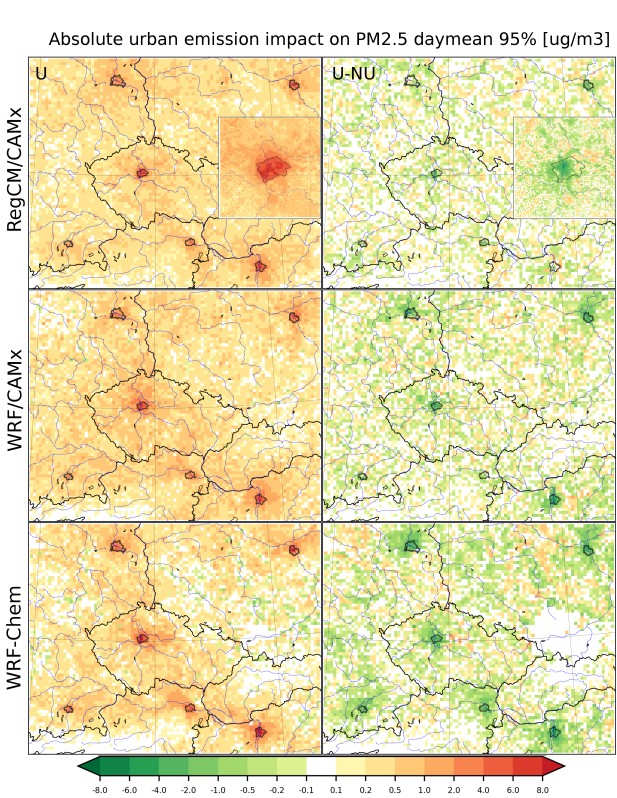

**Figure 12.** Same as Fig. 11 but for PM2.5 and in $\mu gm^{-3}$.



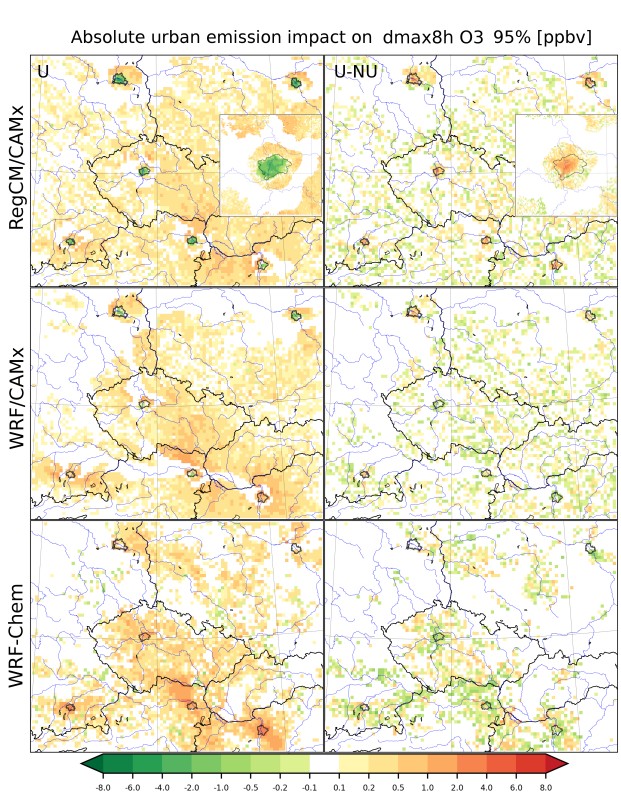

**Figure 13.** Same as Fig. 11 but for the maximum daily 8-hour O$_3$ in ppbv.



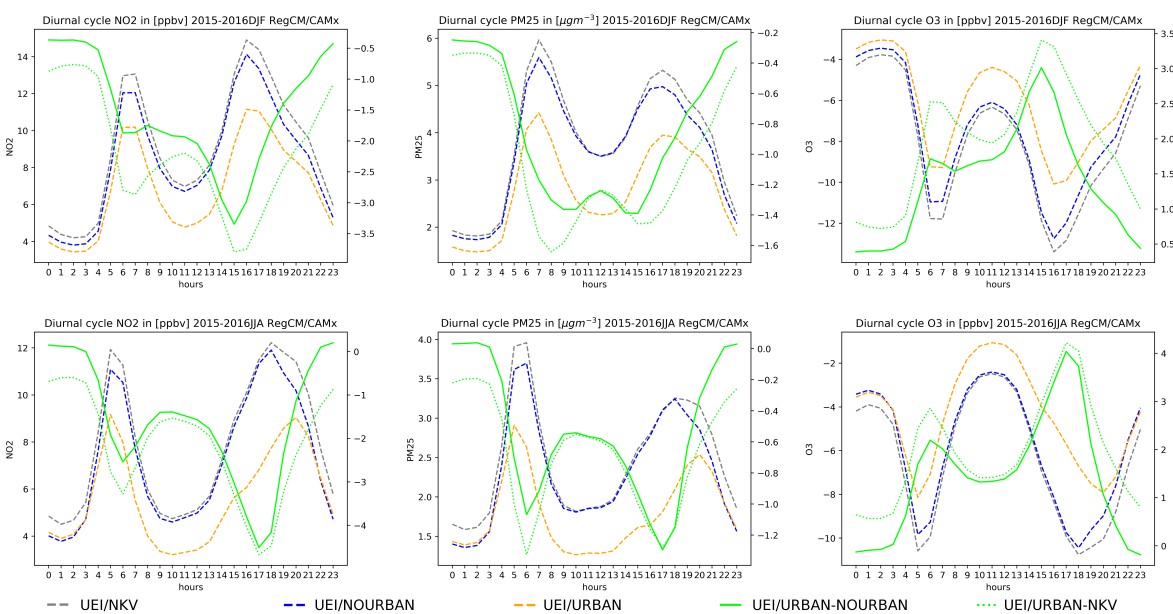

**Figure 14.** Impact of considering the vertical eddy diffusion (Kv) on the urban emission impact (ΔUEI) calculated from RegCM/CAMx simulations for DJF (top) and JJA (bottom) for $NO_2$ (left), PM2.5 (middle) and $O_3$ (right). Grey, blue and green dashed lines stand for the UEI calculated from the "NKV" "NOURBAN" and "URBAN" cases, respectively, and belong to the left y-axis. Green solid and dotted lines denote the ΔUEI as the difference between the "URBAN" and "NOURBAN" and between the "URBAN" and "NKV" cases. Gases in ppbv, PM2.5 in $\mu g m^{-3}$.