# Peer review of "The regional impact of urban emissions on air quality in Europe: the role of the urban canopy effects"

_Atmospheric Chemistry and Physics, 2021_

## Author Comment (AC1)

**Authors response on the Anonymous Referee #2 review of "The regional impact of urban emissions on air quality in Europe: the role of the urban canopy effects"**

We thank anonymous referee #2 for his positive and constructive comments. We will address each of them and our point-by-point responses follow below. All of the editorial/typographical comments will be addressed in the revision. Reviewer's comments are italicized.

*Specific comments/suggestions are listed below.*

*I. Scientific suggestions on the main text*

*General:*

1. *Regarding the choice of cities, the justification is vague; what is to be highlighted for each city?*

   Authors response: During the selection we followed multiple criteria: a) size should be such that at least a few grid-cells in the 9 km domain cover the city, b) cities are sufficiently far from each other to eliminated inter-city influences (see Huszar et al., 2016a), c) the terrain of the city should have minimal variability to eliminate orographical effects, d) cities should be distant from large water bodies and/or sea to eliminate non-symmetric landuse effects around the city (e.g sea breeze effects).
   We admit that the selection could follow a more objective criteria like number of inhabitants or number of sq km-s, however we assumed (and our results showed) that the differences in results between cities are qualitatively very small and the choice of the list of cities has thus very small effect on the results. This has been mentioned also in the manuscript.

2. *Please provide additional explanations about why the focus was on NOx, PM2.5 and O3.*

   Authors' response: NO2 is an abundant pollutant emitted by cities (mainly by transportation) with harmful effects on health when its concentration surpasses a certain threshold value. According to the EEA 2019 Report (EEA, 2019), still 16 of 27 EU members states recorded concentrations above the annual limit value where the cities we analyzed belong too. The reason for choosing PM2.5 is similar: here, Czechia and Poland reported still very large values making PM2.5 a concern. Lastly, O3 is of interest due to its non-linear response to emission changes and, of course also due to its health treat, while its limit values are still being exceeded in many EU member countries.
   The choice of these pollutants was further motivated by the fact, that our analysis is based on our previous works (Huszar et al., 2020a,b; Huszar et al., 2016a) which focused on these three pollutants too (the motivation was the same).

3. *Please provide details about policy relevance of this work.*

   Authors' response: Chemical Transport Models are known to be an essential tools to evaluate policy related air quality measures to control urban air pollution. As such, it is important that all physical processes driving transport of emitted and secondary formed species from urban emissions are correctly represented. Here we show the error the model can produce if the urban canopy meteorological effects are not included. The policies mitigating the pollution treat of urbanization should thus consider of such effects. Therefor we think this study has very high policy relevance. We added these arguments into the revised text.

4. *Revise the term "fingerprint".*

Authors' response: We decided to change this term to „footprint" which is much more common in studies dealing with urban emissions an in general the environmental impact of human activities.

*Specific comments:*

*Page 4, 118-121, harmonise the info in this paragraph with the info provided in the abstract.*

Authors' response: we modified the abstract in accordance with the indicated paragraph.

*Page 6, 191, please summarise the experimental design e.g. in a table.*

Authors' response: Table 1 on Page 27 (in the ACPD version of the manuscript, we placed all the tables and figures at the end of the text after References) provides the summary of all experiments including the information weather a) urban canopy effects b) urban emissions were considered.

*Page 8, 250, please provide more details about model validation, for ozone in particular.*

Authors' response: In the corresponding section ("Model validation") we added more quantitative information on the model biases for all analyzed pollutant to allow the reader to have better idea of the model deficiencies and, in general to make the text more self-consistent.

*Page 9, 255, please provide the name of emission model.*

Authors' response: provided.

*Page 13, 399, DJF ozone is missing, please provide an explanation.*

Authors' response: both NO2 and PM2.5 can be high during all seasons, in particular they can be high in DJF due to limited mixing and stagnant conditions with low PBL. In contrary, O3 is very low in winter due to limited sunshine and high titration by NO, therefor we decided to not consider DJF ozone values in our analysis. Of course, there is an impact on the DJF ozone results, but these have a low relevance due to otherwise very absolute low values.

*II. Typing errors*

Authors' response: all typing errors listed below are corrected in the revised manuscript.

*Page 3, 59, replace "ammonia(" with "ammonia (" - ok*

*Page 6, 167, replace ",the" with ", the" - ok*

*Page 8, 242, replace "models" with "model" - ok*

*Page 11, 341, replace "therefor" with "therefore" - ok*

*Page 11, 353, replace "and4" with "and 4" - ok*

*Page 11, 356, replace "PM" with "PM2.5" - ok*

*Page 12, 363, replace "and6" with "and 6" - ok*

*Page 12, 366, replace "maxima" with "maximum" - ok*

*Page 13, 399, replace Fig/ 7" with "Fig. 7" - ok*

*Page 14, 455, delete the "." from "at.." - ok*

*Page 16, 499, replace "depend" with "depends" - ok*

*Page 16, 520, replace "and" with "an" - ok*

*Page 18, 563, "." is missing - ok*

References:

EEA, 2019: European Environment Agency: Air quality in Europe — 2019 report, EEA Report No 10/2019, doi:10.2800/822355, 2019.

Huszar, P., Belda, M., and Halenka, T.: On the long-term impact of emissions from central European cities on regional air quality, Atmos. Chem. Phys., 16, 1331--1352, doi:10.5194/acp-16-1331-2016, 2016a.

Huszar, P., Karlický, J., Ďoubalová, J., Šindelářová, K., Nováková, T., Belda, M., Halenka, T., Žák, M, and Pišoft, P.: Urban canopy meteorological forcing and its impact on ozone and PM2.5: role of vertical turbulent transport, Atmos. Chem. Phys., 20, 1977-2016, https://doi.org/10.5194/acp-20-11977-2020, 2020a.

Huszar, P., Karlický, J., Ďoubalová, J., Nováková, T., Šindelářová, K., Švábik, F., Belda, M., Halenka, T., and Žák, M.: The impact of urban land-surface on extreme air pollution over central Europe, Atmos. Chem. Phys., 20, 11655–11681, https://doi.org/10.5194/acp-20-11655-2020, 2020b

---

## Author Comment (AC2)

**Authors response on the Anonymous Referee #3 review of "The regional impact of urban emissions on air quality in Europe: the role of the urban canopy effects"**

We thank anonymous referee #3 for his positive and constructive comments. We will address them one-byone and our point-by-point responses follow below. All of the editorial/typographical comments will be addressed in the revision. Reviewer's comments are italicized.

*General comments:*

*Some basic information on the selected cities should be provided, e.g. area, population, population density, main emission sources, etc. What are the differences/similarities between the cities?*

Authors response: We provided information on the criteria for the city selection as well as a table (Tab.2.) containing some basic information about the size/population of the cities. Some information on the most important emissions from these cities is added in the text too. All these information show that the analyzed cities are very similar in terms of size/population, but the share of the main emission categories/sectors is different.

*Chapter 3.1. The validation part should be extended. Some measures of the model performance evaluation should be added. The Authors said that the full validation is described in Huszar et al. (2020b), however that paper gives the validation only for 4 out of 6 analyzed cities. The source (database) of the measurements and type of the stations used for the validation should be also provided.*

Authors response: We indeed admit that an explicit validation for each city should be added and relying on that done for only one of them is not sufficient. We therefor added a comparison of the modelled and measured monthly means of the analyzed species pollutants for each of the city based on all available urban background AirBase station (new Figure.3). Now, it is clear stated in the text what type of AirBase station is used.

Authors further implemented all specific comments on corrections listed below (mainly replacement of text with correction and removal of unnecessary text).

*Specific comments:*

*Line 1: replace „air-quality" by „air quality"*

*Line 2: replace „scales" by „scale"*

*Line 9: replace „rural one while" by „rural one, while"*

*Line 15: replace „In case of" by „In the case of"*

*Line 17: replace „air-pollution" by „air pollution"*

*Line 36: replace „Zha et al., 2019) while for turbulence (especially the vertical eddy diffusivity), strong" by „Zha et al., 2019), while for turbulence (especially the vertical eddy diffusivity) a strong"*

*Line 39: replace „scales" by „scale"*

*Line 40: replace „ones" by „one"*

*Line 45: replace „as well and" by „as well as"*

*Line 57: correct „et al., 20110(@)."*

*Line 58: please use subscript in „PNO3"*

*Lines 59-61: replace „Emissions of ammonia(NH3), although not emitted largely by cities, are an efficient contributor to formation of sulfate and nitrate aerosol (by forming ammonium-sulfates and ammonium-nitrates) and its importance in connection with city emissions is highlighted by many (e.g. Behera and Sharma, 2010, and references therein)."*

*by*

*„Ammonia (NH3), although not emitted largely by cities, is an efficient contributor to formation of sulfate and nitrate aerosol (by forming ammonium-sulfates and ammonium-nitrates) and its importance in connection with city emissions is highlighted in many studies (e.g. Behera and Sharma, 2010, and references therein)."*

*Lines 69-73: replace „On global scale, e.g. Lawrence et al. (2007), Butler and Lawrence (2009), Folberth et al. (2010) or Stock et al. (2013) estimated urban emissions impact, while on regional scales, many studies focused on agglomerations in southern Europe (e.g. Im et al., 2011a, b; Im and Kanakidou, 2012; Finardi et al., 2014), but focused also on other important urban centers like Paris (Skyllakou et al., 2014; Markakis et al., 2015) or London (Hodneborg et al., 2011; Hood et al., 2018)." by*

*„On a global scale, the urban emissions impact was estimated by e.g. Lawrence et al. (2007), Butler and Lawrence (2009), Folberth et al. (2010) or Stock et al. (2013), while on regional scales, many studies focused on agglomerations in southern Europe (e.g. Im et al., 2011a, b; Im and Kanakidou, 2012; Finardi et al., 2014), but also on other important urban centers like Paris (Skyllakou et al., 2014; Markakis et al., 2015) or London (Hodneborg et al., 2011; Hood et al., 2018)."*

*Line 77: replace „available while" by „available, while"*

*Line 80: replace „CO as tracer recently in (Panagi et al., 2020)." by „CO as a tracer recently by Panagi et al. (2020)."*

*Lines 88-89: replace „particle matter (PM) while" by „particulate matter (PM), while"*

*Lines 99-100: remove „into account"*

*Lines 101-102: replace „Here we propose a study that connects the two aspects" by „In this study we propose the combination of the two aspects"*

*Line 107: replace „will be dedicated" by „will be paid"*

*Line 108: replace „(see e.g. Huszar et al. (2020a))." by „(see e.g. Huszar et al., 2020a)."*

*Line 109: replace „particle matter" by „particulate matter"*

*Line 109: replace „then" by „than"*

*Line 118: replace „Two models regional climate models" by „Two regional climate models"*

*Line 129: replace „based on (Oleson et al., 2008)." by „based on Oleson et al. (2008)."*

*Line 131: replace „(Chen and Sun, 2002, PLIN;)" by „(PLIN; Chen and Sun, 2002)"*

*Lines 132-133: replace „(SLUCM; (Kusaka et al., 2001))" by „(SLUCM; Kusaka et al., 2001)"*

*Lines 134-135: replace „(RADM2; Stockwell et al. (1990, 2011))" by „(RADM2; Stockwell et al., 1990; 2011)"*

*Line 136: replace „(MADE/SORGAM; Schell et al. (2001))" by „(MADE/SORGAM; Schell et al., 2001)"*

*Line 145: replace „in to" by „into"*

*Lines 146-147: replace „CAMx code http://www.camx.com/download/support-software.aspx was used for WRF data while"*

*by*

*„CAMx code (http://www.camx.com/download/support-software.aspx) was used for WRF data, while"*

*Line 148: replace „eddy diffusion" by „eddy-diffusion"*

*Line 154: replace „9km, 3km and 1km resolution" by „9 km, 3 km and 1 km resolution"*

*Line 167: replace „simulations,the" by „simulations, the"*

*Line 170: The parenthese is empty*

*Line 172: replace „considered while" by „considered, while"*

*Line 175: replace „The fulfill the goal" by „To fulfill the goal"*

*Line 175: replace „have to performed" by „have been performed"*

*Line 180: replace „Tab.1" by „Table 1"*

*Line 186: replace „considered while" by „considered, while"*

*Line 189: replace „eddy diffusion" by „eddy-diffusion"*

*Line 208: replace „republic" by „Republic"*

*Line 211: replace „nitrogen(NOx)" by „nitrogen (NOx)"*

*Line 240: replace „the regional climate" by „RCM"*

*Line 240: replace „Tab. 1" by „Table 1"*

*Line 242: replace „models" by „model"*

*Line 253: replace „ozone while" by „ozone, while"*

*Line 261: replace „expect" by „except"*

*Line 273: replace „impact shown in the first one while" by „impact is shown in the first one, while"*

*Line 285: replace „not confined" by „not limited"*

*Line 290: replace „(seen in Huszar et al. (2020b))" by „(see Huszar et al., 2020b)"*

*Line 294: replace „then" by „than"*

*Line 297: replace „then" by „than"*

*Line 298: replace „then" by „than"*

*Line 301: replace „For PM2.5 again," by „For PM2.5 (Fig. 5) again,"*

*Line 301: replace „NO2.The" by „NO2. The"*

*Line 302: replace „0.5μgm−3" by „0.5 μgm−3"*

*Line 304: replace „6μgm−3" by „6 μgm−3"*

*Line 310: replace „During JJA," by „During JJA (Fig. 6),"*

*Line 316: replace „on regional ozone" by „on regional ozone (Fig. 7)"*

*Line 318: replace „In case CAMx" by „In case of CAMx"*

*Line 325: replace „due to urban" by „due to the urban"*

*Line 332: replace „ozone while" by „ozone, while"*

*Line 340: replace „hours while" by „hours, while"*

*Lines 340-341: replace „different of urban emission impact" by „different impact of urban emission"*

*Line 341: replace „therefor" by „therefore"*

*Line 348: replace „respectively while" by „respectively, while"*

*Line 351: replace „respectively while" by „respectively, while"*

*Line 353: replace „and4." by „and 4."*

*Line 354: replace „models" by „model"*

*Line 361: replace „respectively while" by „respectively, while"*

*Line 363: replace „and6." by „and 6."*

*Line 365: replace „In case of ozone," by „In case of ozone (Fig. 10),"*

*Line 369: replace „cases while" by „cases, while"*

*Line 375: replace „WRF-Chem) while in JJA, the" by „WRF-Chem), while in JJA the"*

*Line 385: replace „rule" by „role"*

*Line 395: replace „ozone, Fig. 13," by „ozone (Fig. 13),"*

*Line 399: replace „Fig/ 7." by „Fig. 7."*

*Line 409: replace „are considered" by „is considered"*

*Line 420: replace „case." by „case are presented."*

*Line 427: replace „windspeeds" by „wind speed"*

*Line 431: replace „with highest" by „with the highest"*

*Line 440: replace „air-quality" by „air quality"*

*Line 446: replace „to measurements" by „to the measurements"*

*Line 455: replace „by (Nopmongcol et al., 2012)." by „by Nopmongcol et al. (2012)."*

*Line 491: replace „eddy diffusion" by „eddy-diffusion"*

*Line 491: replace „wind-speeds" by „wind speed"*

*Line 520: replace „and important" by „an important"*

*Line 521: replace „This caused" by „This is caused"*

*Line 524: replace „are largest" by „are the largest"*

*Line 524: replace „wind-speed" by „wind speed"*

*Line 526: replace „During that the" by „During that time the"*

*Line 531: replace „eddy diffusion" by „eddy-diffusion"*

*Line 533: replace „see (Karlický et al., 2020))." by „see Karlický et al., 2020)."*

*Line 535: replace „their" by „the"*

*Line 538: replace „wind-speed" by „wind speed"*

*Line 539: replace „the the modulation" by „the modulation"*

*Page 29, Fig. 3 caption: replace „five selected cities" by „six selected cities"*

*Page 35, Fig. 11 caption: replace „The urban emission impact (UEI) of selected cities" by „The impact of urban emission (UEI) for selected cities"*

*Page 38, Fig. 14 caption: replace „eddy diffusion" by „eddy-diffusion"*